# Generating Origin-Destination Matrices in Neural Spatial Interaction Models

**Ioannis Zachos**[1*]     **Mark Girolami**[1,2]     **Theodoros Damoulas**[2,3]

[1]Department of Engineering, Cambridge University, Cambridge, CB2 1PZ.
[2]The Alan Turing Institute, London, NW1 2DB.
[3]Departments of Statistics & Computer Science, University of Warwick, Coventry, CV4 7AL.
`iz230@cam.ac.uk`

## Abstract

Agent-based models (ABMs) are proliferating as decision-making tools across policy areas in transportation, economics, and epidemiology. In these models, a central object of interest is the discrete origin-destination matrix which captures spatial interactions and agent trip counts between locations. Existing approaches resort to continuous approximations of this matrix and subsequent ad-hoc discretisations in order to perform ABM simulation and calibration. This impedes conditioning on partially observed summary statistics, fails to explore the multimodal matrix distribution over a discrete combinatorial support, and incurs discretisation errors. To address these challenges, we introduce a computationally efficient framework that scales linearly with the number of origin-destination pairs, operates directly on the discrete combinatorial space, and learns the agents' trip intensity through a neural differential equation that embeds spatial interactions. Our approach outperforms the prior art in terms of reconstruction error and ground truth matrix coverage, at a fraction of the computational cost. We demonstrate these benefits in large-scale spatial mobility ABMs in Cambridge, UK and Washington, DC, USA.

## 1   Introduction

High-resolution complex simulators such as agent-based models (ABMs) are increasingly deployed to assist policymaking in transportation [10, 21], social sciences [3, 8, 14, 34], and epidemiology [16, 22]. They simulate individual agent interactions governed by stochastic dynamic systems, giving rise to an aggregate, in a mean field sense, continuous emergent structure. This is achieved by computationally expensive forward simulations, which hinders ABM parameter calibration and large-scale testing of multiple policy scenarios [25]. Considering ABMs for the COVID-19 pandemic [16] as an example, the continuous mean field process corresponds to the spatial intensity of the infections which is noisily observed at some spatial aggregation level, while the individual and discrete human contact interactions that give rise to that intensity are at best partially observed or fully latent. In transportation and mobility, running examples in this work, the continuous mean field process corresponds to the spatial intensity of trips arising from unobserved individual agent trips between discrete sets of origin and destination locations [19, 10].

The formal object of interest that describes the discrete count of these spatial interactions, e.g. agent trips between locations, is the origin-destination matrix (ODM). It is an $I \times J$ (two-way) contingency table $\mathbf{T}$ with elements $T_{i,j} \in \mathbb{N}$ counting the interactions of two spatial categorical variables $i, j \in \mathbb{N}_{>0}$, see Fig. 1. It is typically sparse due to infeasible links between origin and destination locations, and partially observed through summary statistics – such as table row and/or column marginals – due to privacy concerns, data availability, and data collection costs. Operating at

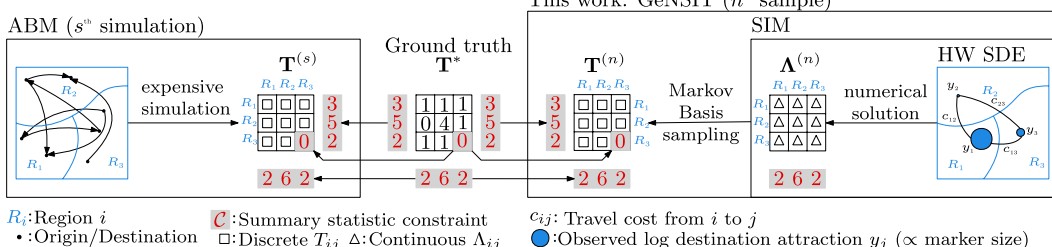

R_i:Region i    $\mathcal{C}$:Summary statistic constraint    $c_{ij}$: Travel cost from $i$ to $j$
•:Origin/Destination    □:Discrete $T_{ij}$  △:Continuous $\Lambda_{ij}$    🔵:Observed log destination attraction $y_j$ ($\propto$ marker size)

Figure 1: The ground truth discrete ODM (two-way contingency table) can be reconstructed through either multiple expensive ABM simulations [2] (ABM rectangle) or approximated by a continuous representation $\Lambda$ coupled with the Harris-Wilson SDE (SIM rectangle). In the latter, the ground truth can be reconstructed by sampling in the discrete combinatorial space of constrained ODMs conditioned on $\Lambda$ (GENSIT rectangle). ABM simulations scale with $\mathcal{O}(M \log(M))$ compared to GENSIT which scales with $\mathcal{O}(IJ)$, where $M \gg I + J$ is the size of the agent interaction graph.

the discrete ODM level and learning this latent contingency table from summary statistics is vital for handling high-resolution spatial constraints and partial observations such as the total number of agents interacting between a pair of locations. It is also necessary for *population synthesis* in ABMs [15], which is performed prior to simulation in order to reduce the size of the ABM's parameter space. Moreover, it avoids errors and biases due to ad-hoc discretisation required when working with continuous approximations of the underlying discrete ODM $\mathbf{T}^*$.

Traditional ABMs, Fig. 1 (Left), simulate individual-level and spatially granular discrete ODMs at a high computational cost [2], which scales at least with $\mathcal{O}(M \log(M))$, where $M \gg I + J$ is the size of the agent interaction graph. These ODMs are then aggregated using a sum pooling operation to regional ones, $\mathbf{T}^{(s)}$, whose summary statistics $\mathcal{C}$ correspond to observed data.

However, the lower-dimensional subspace of contingency tables $\mathbf{T}$ satisfying constraints $\mathcal{C}$ (e.g. row and column marginals), denoted by $\mathcal{T}_{\mathcal{C}}$, is known to be combinatorially large [11] and therefore sampling and optimisation in that space is notoriously hard since it requires enumerating all elements of $\mathcal{T}_{\mathcal{C}}$. This underlying challenge is the reason why prior art [35, 13, 41, 32, 27, 17] has been imposing a continuous relaxation of the ODM estimation problem, where the continuous approximation of the discrete contingency table restricts inference at the agent trip intensity level $\Lambda$. This leads to quantisations and inefficient rejection sampling, see Fig. 2.

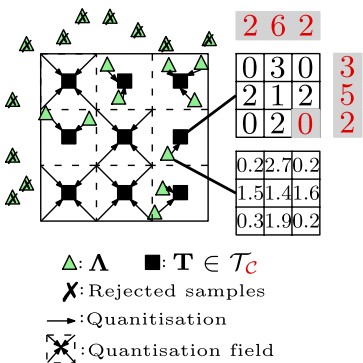

△: $\Lambda$    ■: $\mathbf{T} \in \mathcal{T}_{\mathcal{C}}$
✗:Rejected samples
→:Quanitisation
✗:Quantisation field

Figure 2: The space $\mathcal{T}_{\mathcal{C}}$ of $3 \times 3$ discrete ODMs with summary statistics $\mathcal{C}_T$. Sampling on the continuous relaxation of $\mathcal{T}_{\mathcal{C}}$ ($\Lambda$ level) with quantisation can lead to either large rejection rates, or poor exploration of the distribution over $\mathcal{T}_{\mathcal{C}}$.

Such intensity-level Spatial Interaction Models (SIMs) [45, 31], Fig. 1 (Right), are derived from entropy maximisation arguments, Sec. 2, with summary statistics constraints. These models are embedded in the Harris-Wilson (HW) system of differential equations [20] that describe the time evolution of location attraction and reflects the utility gained from reaching a destination. Coupling SIMs with the HW model introduces an inductive bias that regularises the continuous ODM and facilitates a mean-field ABM approximation. This approximation effectively acts as a cheap ABM surrogate or emulator, facilitating faster forward simulations to be run by practitioners [4]. This has tangible benefits to ABM calibration allowing faster exploration of the parameter space. Our enhanced ODM reconstruction error demonstrates GeNSIT's ability to sufficiently approximate the ABM simulator at a fraction of the computational cost. However, if both the continuous ODM row and column marginals are fixed, the HW model becomes redundant and the SIM can either be greedily approximated through iterative proportional fitting [45] which is sensitive to initialisation or become unidentifiable as the number of parameters scales with both the number of origins and destinations. Furthermore, conditioning on individual continuous ODM cells creates discontinuities as the SIM becomes a piecewise function, also preventing its coupling into the HW model.

Competing approaches that operate directly on the discrete ODM space and are motivated by econometric arguments are discrete choice models [44]. However, these cannot encode summary statistic constraints without introducing large rejection rates. The work of [9] leverages SIMs to sample discrete ODMs but removes intensity constraints through log-linearity assumptions and does not exploit the physics structure, effectively stripping SIMs of their advantages over other choice models. Markov Chain Monte Carlo routines have been devised to address these issues by learning the SIM parameters [13] and the associated discrete ODM table over its entire support [47]. Such routines incur a computational overhead in the order of at least $I$ or $J$ due to the intractability of the Harris-Wilson model prior, rendering them prohibitive for large-scale applications. Neural Network (NN) parameter calibration of SIMs has been empirically shown to achieve up to ten-fold speed-ups during training [17] by using a numerical discretisation scheme for the Harris-Wilson model as a forward solver. Despite the significant advantages offered by NNs, they operate strictly in the continuous intensity level and cannot generate discrete agent-level ODMs.

In this paper, we introduce a computationally scalable framework named **Ge**nerating **N**eural **S**patial **I**nteraction **T**ables (GENSIT) for exploring the constrained discrete ODM space using closed-form or Gibbs Markov Basis sampling while neurally calibrating the underlying physics-driven SIM parameters of its continuous representation, as shown in Fig. 3. Our framework scales linearly with the number of origin-destination pairs $IJ$, which is at least $I$ or $J$ times faster than MCMC [13, 47]. We offer faster ODM reconstruction and enhanced uncertainty quantification compared to continuous approaches in seminal works [13, 17] and hybrid (discrete and continuous) approaches [47] in terms of the number of iterations required. It is the first framework that jointly explores the constrained continuous and discrete combinatorial ODM spaces in linear in the number of origin-destination pairs time. It does so while outperforming the prior art in terms of reconstruction error and ground truth table coverage while enabling the integration of a broader set of constraints, if these are available.

Our framework has merit beyond ODM sampling in ABMs. The challenge of learning discrete contingency tables constrained by their summary statistics extends to other fields. Contingency tables have been widely studied in multiple instance learning [33, 12, 48] and ecological inference [37, 36, 39]. In Neuroscience one estimates the efficiency, cost and resilience (equivalent to $T_{ij}$) of neural pathways between pairs of brain regions $(i, j)$ to understand communication and information processing [38],[18]. Epidemiology also investigates social contact matrices quantifying the number of contacts ($T_{ij}$) between types of individuals $(i, j)$ stratified by demographics, such as age [29].

## 2    Spatial Interaction Intensities and Contingency Tables

Consider $A$ agents travelling from $I$ residences (origins) to $J$ workplaces (destinations). The expected number of trips (intensity) between origin $i$ and destination $j$ is $\Lambda_{ij}$ and is unobserved. The average number of agents starting (ending) their journey from each origin $i$ (to each destination $j$) is:

$$\Lambda_{i+} = \sum_{j=1}^{J} \Lambda_{ij}, \quad i = 1, \ldots, I, \qquad \Lambda_{+j} = \sum_{i=1}^{I} \Lambda_{ij}, \quad j = 1, \ldots, J. \tag{1}$$

The expected total number of agents travelling is assumed conserved:

$$\Lambda_{++} = \sum_{i=1}^{I} \Lambda_{i+} = \sum_{j=1}^{J} \Lambda_{+j} = A. \tag{2}$$

The family of models for intensities $\boldsymbol{\Lambda}$ that assimilate any collection of the above constraints are called Spatial Interaction Models [45]. The demand for each destination depends on its attractiveness denoted by $\mathbf{z} := (z_1, \ldots, z_J) \in \mathbb{R}_{>0}^J$. In our example, this is the number of jobs available at each destination. Let the log-attraction be $\mathbf{x} := \log(\mathbf{z})$. Between two destinations of similar attractiveness, agents are assumed to prefer nearby destinations. Therefore, a cost matrix $\mathbf{C} = (c_{i,j})_{i,j=1}^{I,J}$ is introduced to reflect travel impedance. These assumptions are justified by economic arguments [31] and establish the basis for the agents' utility function. The maximum entropy distribution of agent trips subject to $\Lambda_{++} = A$ yields a totally constrained SIM intensity:

$$\Lambda_{ij} = \frac{\Lambda_{++} \exp(\alpha x_j - \beta_{ij})}{\sum_{k,m}^{I,J} \exp(\alpha x_m - \beta c_{km})}, \tag{3}$$

where $\mathcal{C}_\Lambda = \{\Lambda_{++}\}$ is the set of summary statistic constraints on $\Lambda$. Henceforth, we set $\Lambda_{++} \in \mathcal{C}_\Lambda$ unless otherwise stated. The vector $\boldsymbol{\theta} = (\alpha, \beta)$ contains the agents' two utility parameters controlling the effects of attractiveness and deterrence on the expected number of trips $\Lambda$. If $\alpha$ grows larger relative to $\beta$ then agents gravitate towards destinations with higher job availability regardless of the travel cost incurred, and vice versa. Further, if we also fix the expected origin demand $\Lambda_{\cdot+}$, then we obtain the following singly (also known as production) constrained SIM intensity:

$$\Lambda_{ij} = \frac{\Lambda_{i+} \exp(\alpha x_j - \beta c_{ij})}{\sum_m^J \exp(\alpha x_m - \beta c_{im})}, \tag{4}$$

where $\mathcal{C}_\Lambda$ is expanded to include $\Lambda_{\cdot+}$ in this case. Moreover, setting $\mathcal{C}_\Lambda = \{\Lambda_{++}, \Lambda_{\cdot+}, \Lambda_{+\cdot}\}$ yields a doubly constrained SIM

$$\Lambda_{ij} = \Lambda_{i+}\Lambda_{+j} \exp(\alpha x_j - \beta c_{ij})O(i)D(j), \tag{5}$$

where $O(i), D(j)$ are called balancing factors that ensure that $\mathcal{C}_\Lambda$ are satisfied. The balancing factors introduce $I + J$ unknown parameters, rendering the intensity model unidentifiable. Alternatively, these factors are approximately recursively using iterative proportional fitting [45], which is sensitive to initialisation. Including individual cell constraints in $\mathcal{C}_\Lambda$ breaks the continuity of $\Lambda$ as a function of its parameters. For these reasons, the doubly and/or cell-constrained SIMs are prohibitive for use in statistical inference. See App. B.1.1 for more information on SIMs as a modelling choice.

We note that additional data at the origin, destination and origin-destination level can be assimilated into SIMs. This can be achieved by incorporating them as terms in the maximum entropy argument used to derive the $\Lambda$ functional forms in equations (3), (4), and (5). We note that the SIM's $\Lambda$ is equivalent to the multinomial logit [30], which generalises our $\Lambda$ construction to accommodate for more data. See App. B.1.2 for a guide on eliciting agent utility functions.

It has been shown that the destination attractiveness $\mathbf{z} = \exp(\mathbf{x})$ in families of SIMs is governed by the Harris-Wilson system of $J$ coupled stochastic differential equations (SDEs) [20, 13]:

$$\frac{\mathrm{d}z_j}{\mathrm{d}t} = \epsilon z_j(\Lambda_{+j} - \kappa z_j + \delta) + \sigma z_j \circ B_{j,t}, \quad \mathbf{z}(0) = \mathbf{z}' \tag{6}$$

where $\epsilon > 0$ is a responsiveness parameter, $\kappa > 0$ is the number of agents competing for one job, $\delta \geq 0$ is a parameter related to the smallest number of jobs a destination can have, $\sigma > 0$ is the standard deviation of SDE's noise, and $\mathbf{B}_t$ is a $J$-dimensional Wiener process. The term $\Lambda_{+j} - \kappa z_j + \delta$ in (6) reflects the net job capacity at destination $j$. If more agents travel to $j$ than there are jobs there (positive capacity), this may signify a boost in $j$'s local economy, which would trigger a rise in job availability, and vice versa. The diffusion term stochastically perturbs this trend to account for unobserved events, such as local government interventions affecting employment. By the HW-SDE, the SIM intensity is a stochastic and physics-driven quantity reflecting the agents' average number of trips $\Lambda$ between their residences and workplaces. However, the SIM intensity differs from the realised number of trips $\mathbf{T}$ agents make. The two notions are connected as follows:

$$T_{ij}|\Lambda_{ij} \sim \text{Poisson}(\Lambda_{ij}), \tag{7}$$

$\mathbf{T}$ is the $I \times J$ discrete contingency table summarising the number of agents travelling from $i$ to $j$, and $T_{ij} \perp T_{i'j'}|\Lambda_{ij}, \Lambda_{i'j'} \; \forall \; i \neq i', j \neq j'$. Any choice of model for $T_{ij}|\Lambda_{ij}$ (say Poisson or Binomial) becomes equivalent upon conditioning on sufficient summary statistics $\mathcal{C}_T$ [1]. We note that the conditional independence of the $T_{ij}$'s given the $\Lambda_{ij}$'s and that $\mathbf{T}$ inherits all constraints from $\Lambda$ such that every summary statistic constraint in $\Lambda$-space is also applied in $\mathbf{T}$-space, i.e. we always set $\Lambda_{++} = \mathbb{E}[T_{++}|\mathcal{C}_T]$ and $\Lambda_{i+} = \mathbb{E}[T_{i+}|\mathcal{C}_T]$. The hard-coded constraints $\mathcal{C}_T$ are realisations of the Poisson random variables $T_{++}|\Lambda, T_{i+}|\Lambda, T_{+j}|\Lambda$, and therefore are no longer random. They can be thought of as noise-free data on the discrete table space. Following the notational convention for $\Lambda$, we define $T_{i+} = \sum_{j=1}^J T_{ij}$, $T_{+j} = \sum_{i=1}^I T_{ij}$ and $T_{++} = \sum_{i,j=1}^{I,J} T_{ij}$, respectively. The aforementioned summary statistics become Dirac random variables upon conditioning on $\mathbf{T}$. The union of table and intensity constraints is summarised by $\mathcal{C}$. We sometimes drop subscripts from $\mathcal{C}$ for clarity. See App. A for more information on the notation used.

## 3 Neural Calibration of Spatial Interaction Models

We now introduce our framework[1] (GENSIT). We estimate the parameters of the continuous SIM intensity by employing an ensemble of Neural Networks $\psi_{NN} : \mathbb{R}^J \to \mathbb{R}^2$. This allows us to bypass

---

[1]Codebase found at https://github.com/YannisZa/GeNSIT

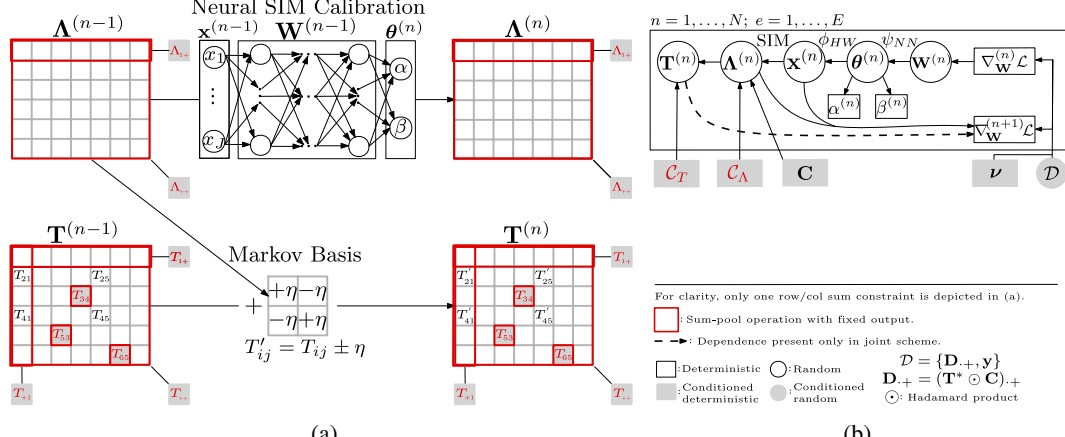

Figure 3: GENSIT: (a) successive iterations of Alg. 1 for a given ensemble member, (b) plate diagram for every iteration, ensemble member. We propose two schemes: a Joint and a Disjoint (see App. B.3.1 for details). Contrary to the latter, the former passes table $\mathbf{T}$ information to the loss $\mathcal{L}$ (see $\text{-}\!\rightarrow$ in (b)). We perform an optimisation step in the intensity $\mathbf{\Lambda}$ space and a sampling step in $\mathbf{T}$ space, with associated complexities $\mathcal{O}(\tau J + IJ)$ and $\mathcal{O}(IJ)$. The $\mathbf{\Lambda}$ arises by the well-known family of SIMs (3),(4) coupled with the HW-SDE (6). The $\mathbf{T}$ sampling step generates discrete $\mathcal{C}_T$-constrained ODMs contrary to [13, 17], which only operate on the continuous mean-field level $\mathbf{\Lambda}$.

the computational challenges of solving the HW-SDE inverse problem within a Bayesian framework [13, 47]. Conditioned on those estimates and for every ensemble member $e = 1, \ldots, E$, we solve the HW-SDE to get estimates of the time-evolved log destination attraction $\hat{\mathbf{x}}$ after $\tau$ time steps using an Euler-Maruyama numerical solver $\phi_{HW} : \mathbb{R}^J \rightarrow \mathbb{R}^J$ [28]. This allows us to incorporate the HW physics model into our parameter estimation without sampling from the SDE's intractable steady-state Boltzmann-Gibbs distribution, which was the case in [13, 47].

Instead of a log-destination attraction data ($\mathbf{y}$) likelihood, we compute a loss operator $\mathcal{L}(\cdot \; ; \; \mathcal{D}, \boldsymbol{\nu})$ that can assimilate data $\mathcal{D}$ from multiple sources on any transformation of $\hat{\mathbf{T}}, \hat{\mathbf{\Lambda}}, \hat{\mathbf{x}}$. We note that $\boldsymbol{\nu}$ is used to denote the loss hyperparameters (see Tab. 5). Then, the NN parameters (weights and biases) $\mathbf{W}$ are updated using back-propagation using a suite of optimisation algorithms [23]. This step requires derivatives of the loss with respect to the NN parameters $\nabla_{\mathbf{W}} \mathcal{L}(\cdot \; ; \; \mathcal{D}, \boldsymbol{\nu})$ to be computed, which is achieved using off-the-shelf auto-differentiation libraries [26]. This gradient is informed by $\hat{\mathbf{x}}$ estimates and therefore by the dynamics of the HW-SDE in (6). Hence, the resulting SIM intensity is both stochastic and physics-driven. These steps are depicted in Fig. 3b by following the arrows from right to left and in steps 6 to 12 of Alg. 1. We note that our framework is divided into two sampling schemes: a *Joint* and a *Disjoint*. The former passes $\mathbf{T}$ information to the loss operator $\mathcal{L}$, whereas the latter does not. The Joint scheme corresponds to a Gibbs sampler on the full posterior marginals $\boldsymbol{\theta}|(\mathbf{x}, \mathbf{T}, \mathcal{C}, \mathcal{D})$, $\mathbf{x}|(\boldsymbol{\theta}, \mathbf{T}, \mathcal{C}, \mathcal{D})$ and $\mathbf{T}|(\boldsymbol{\theta}, \mathbf{x}, \mathcal{C}, \mathcal{D})$. The Disjoint scheme corresponds to a collapsed Gibbs sampler where we sample from $\boldsymbol{\theta}|(\mathbf{x}, \mathcal{C}, \mathcal{D})$, $\mathbf{x}|(\boldsymbol{\theta}, \mathcal{C}, \mathcal{D})$ and then from $\mathbf{T}|(\boldsymbol{\theta}, \mathbf{x}, \mathcal{C}, \mathcal{D})$ by integrating out $\mathbf{T}$ (see App. B.3.1).

The described optimisation routine yields a point estimate for $\mathbf{\Lambda}$ for any given ensemble member $e$ and iteration $n$. We either increase the ensemble size $E$ to obtain a distribution over $\mathbf{\Lambda}$, or we can treat $\mathbf{\Lambda}$ as a deterministic mapping of realisations of random variables $\mathbf{x}, \boldsymbol{\theta}$ whose generalised posterior [7, 24] is:

$$p(\mathbf{x}, \boldsymbol{\theta}|\mathcal{D}) \propto \exp(-\mathcal{L}(\mathbf{x}, \boldsymbol{\theta} \; ; \; \mathcal{D}, \boldsymbol{\nu}))p(\mathbf{x}|\boldsymbol{\theta})p(\boldsymbol{\theta}), \qquad (8)$$

where $\boldsymbol{\nu}$ are loss-related hyperparameters. Each loss evaluation can be treated as a sample from a generalised likelihood (first term), while samples from the $\mathbf{x}$ prior (second term) are obtained by forward solving $\phi_{HW}$. Finally, we can enforce a prior over $\boldsymbol{\theta}$ by appropriately initialising the NN parameters $\mathbf{W}$.

The continuous agent trip intensity $\mathbf{\Lambda}$ (continuous ODM) samples need to be mapped to discrete agent trips $\mathbf{T}$ (discrete ODM), which are required for agent population synthesis. We proceed by

introducing the necessary machinery to achieve this. We wish to sample from the target measure $\mu$ evaluated as $\mu(\mathbf{T}|\boldsymbol{\Lambda}, \mathcal{C})$.

## 3.1 Constrained Table Sampling

Let the table cells be indexed by $\mathcal{X} = \{(i, j) : 1 \leq i \leq I, 1 \leq j \leq J\}$ such that $T(x) = T_{ij}$ is the table value of cell $x = (i, j) \in \mathcal{X}$. All non-negative two-way contingency tables $\mathbf{T}$ are assumed to be members of a discrete space $\mathcal{T}$. The $k$-th basis operator $\mathbb{1}_k : \mathcal{X}_k \to \{0, 1\}^{I+J}$ is defined to be

$$\mathbb{1}_k(x) = (0, \ldots, 0, \underbrace{1}_{\text{entry } i}, 0, \ldots, 0, \underbrace{1}_{\text{entry } I+j}, 0) . \tag{9}$$
$$\underbrace{\qquad\qquad\qquad\qquad\qquad\qquad\qquad}_{I+J \text{ entries}}$$

Hence, we can define summary statistic operators $\mathcal{S}_k : \mathcal{T} \to \mathbb{N}^{I+J}$ as linear combinations of the basis operator, that is $\mathcal{S}_k(\mathbf{T}) = \sum_{x \in \mathcal{X}_k} \mathbf{T}(x) \mathbb{1}_k(x)$. A collection of such summary statistic operators is abbreviated by $\boldsymbol{\mathcal{S}}(\mathbf{T})$.

---

**Algorithm 1** : **Ge**nerating **N**eural **S**patial **I**nteraction **T**ables. $\qquad\qquad \mathcal{O}(NE(\tau J + IJ))$

---

1: **Inputs**: evidence: $\mathbf{C}, \mathcal{C}, \mathcal{D}$, funcs.: $\mathcal{L}, \psi_{NN}, \phi_{HW}, \mathcal{M}, \mu$, hyperparams.: $N, E, \tau, \boldsymbol{\nu}, \kappa, \delta, \sigma, \epsilon$.
2: **Outputs**: $\mathbf{x}^{(1:N)}, \boldsymbol{\theta}^{(1:N)}, \boldsymbol{\Lambda}^{(1:N)}, \mathbf{T}^{(1:N)}$.
3: **for** each ensemble member $e = 1, \ldots, E$ **do**
4: $\qquad$ Initialise $\mathbf{W}^{(0)}, \mathbf{T}^{(0)}$.
5: $\qquad$ **for** each iteration $n = 1, \ldots, N$ **do**
6: $\qquad\qquad \boldsymbol{\theta}^{(n)} \leftarrow \psi_{NN}(\mathbf{y}; \mathbf{W}^{(n-1)})$, with $\mathbf{y} \in \mathcal{D}$. $\qquad\qquad\qquad$ ▷ Forward solving Neural Net.
7: $\qquad\qquad \boldsymbol{\theta}^{(n)} \leftarrow (\boldsymbol{\theta}^{(n)}, \kappa, \delta, \sigma, \epsilon)$.
8: $\qquad\qquad \mathbf{x}_t^{(n)} \leftarrow \mathbf{x}_t^{(n-1)} \ \forall \, t = 1, \ldots, \tau$.
9: $\qquad\qquad$ **for** each time step $t = 1, \ldots, \tau - 1$ **do** $\qquad$ ⎫
10: $\qquad\qquad\qquad \mathbf{x}_{t+1}^{(n)} \leftarrow \phi_{HW}(\mathbf{x}_t^{(n)}, \boldsymbol{\theta}^{(n)})$. $\qquad\qquad$ ⎬ Incorporating $\quad$ ▷ Solving Harris-Wilson SDE.
11: $\qquad\qquad \mathbf{x}^{(n)} \leftarrow \mathbf{x}_\tau^{(n)}$. $\qquad\qquad\qquad\qquad\qquad\qquad$ ⎭ physics.
12: $\qquad\qquad \boldsymbol{\Lambda}^{(n)} \leftarrow \Lambda_{\mathcal{C}}(\mathbf{x}^{(n)}, \boldsymbol{\theta}^{(n)}, \mathbf{C})$ using (3) or (4). $\qquad\qquad$ ▷ Computing SIM intensity.
13: $\qquad\qquad$ **if** Joint sampling scheme is used **then** $\qquad$ ⎫
14: $\qquad\qquad\qquad L^{(n)} \leftarrow \mathcal{L}(\mathbf{x}^{(n)}, \mathbf{T}^{(n-1)}, \boldsymbol{\Lambda}^{(n)} \ ; \ \mathcal{D}, \boldsymbol{\nu})$. $\quad$ ⎬ Computing
15: $\qquad\qquad$ **else** $\qquad\qquad\qquad\qquad\qquad\qquad\qquad\qquad\qquad$ ⎨ Neural Net loss.
16: $\qquad\qquad\qquad L^{(n)} \leftarrow \mathcal{L}(\mathbf{x}^{(n)} \ ; \ \{\mathbf{y}\}, \boldsymbol{\nu})$. $\qquad\qquad\qquad$ ⎬
17: $\qquad\qquad$ Compute $\nabla_{\mathbf{W}}^{(n)} \mathcal{L}$. $\qquad\qquad\qquad\qquad\qquad\qquad$ ⎭
18: $\qquad\qquad$ Update $\mathbf{W}^{(n)}$ using back-propagation. $\qquad$ ▷ Updating Neural Net Weights with SGD.
19: $\qquad\qquad$ **if** $\mu$ is not tractable **then** $\qquad\qquad\qquad$ ⎫
20: $\qquad\qquad\qquad$ Sample $\mathbf{f}_l$ uniformly at random from $\mathcal{M}$. $\quad$ ⎬ Gibbs Markov
21: $\qquad\qquad\qquad$ Find $\text{supp}\{\eta\}$ such that $\mathbf{T}^{(n-1)} + \eta \mathbf{f}_l \geq 0$. ⎨ Basis sampling. $\quad$ ⎫
22: $\qquad\qquad\qquad \mathbf{T}^{(n)} \sim \mu(\mathbf{T}^{(n-1)} + \eta \mathbf{f}_l | \boldsymbol{\Lambda}^{(n)}, \mathcal{C}_T)$. $\quad$ ⎭ $\qquad\qquad$ ⎬ Sampling
23: $\qquad\qquad$ **else** $\qquad\qquad\qquad\qquad\qquad\qquad\qquad\qquad$ ⎫ $\qquad\qquad$ ⎨ discrete ODM.
24: $\qquad\qquad\qquad \mathbf{T}^{(n)} \sim \mu(\cdot \ | \boldsymbol{\Lambda}^{(n)}, \mathcal{C}_T)$. $\qquad\qquad$ ⎬ Closed-form $\qquad$ ⎭
$\qquad\qquad\qquad\qquad\qquad\qquad\qquad\qquad\qquad\qquad\qquad$ ⎭ sampling.

---

Constraints on $\mathbf{T}$ can be expressed as fixed summary statistics $\mathcal{C}_T = \{\mathbf{s}_1, \ldots, \mathbf{s}_K\}$, where each $\mathbf{s}_k$ is a fixed evaluation of $\mathcal{S}_k(\mathbf{T})$ with respect to the basis operator. For example, constraint $\{\mathbf{T}_{\cdot+}, \mathbf{T}_{+\cdot}\}$ can be expressed in terms of the basis operator (9) over the entire cell set $\mathcal{X}$.

**Definition 3.1.** Let $\mathcal{C}_T = \{\mathbf{s}_1, \ldots, \mathbf{s}_K\}$ be a set of constraints based on summary statistics operators $\boldsymbol{\mathcal{S}}$. A table $\mathbf{T}$ is $\mathcal{C}_T$-*admissible* if and only if $\boldsymbol{\mathcal{S}}(\mathbf{T}) = \{\mathbf{s}_k\}_{k=1}^K \in \mathcal{C}_T$.

The subspace of $\mathcal{T}$ containing all $\mathcal{C}_T$-admissible $I \times J$ contingency tables is $\mathcal{T}_{\mathcal{C}} = \{\mathbf{T} \in \mathcal{T} : \boldsymbol{\mathcal{S}}(\mathbf{T}) = \mathcal{C}_T\}$. This space contains all discrete ODMs consistent with the aggregate summary statistics $\mathcal{C}_T$. Our goal is to efficiently sample from a measure $\mu$ on $\mathcal{T}_{\mathcal{C}}$ subject to arbitrary $\mathcal{C}_T$.

The target distribution is tractable, i.e. the entire table can be sampled directly in closed-form, if and only if the universe of summary statistic constraints contains at most one table marginal

(row or col. sum). This covers the cases where $\mathcal{C}_T^{\text{tractable}} \subseteq \mathcal{P}(\{T_{++}, \mathbf{T}_{\cdot+}, \{\mathbf{T}_{\mathcal{X}_l} | \mathcal{X}_l \subseteq \mathcal{X}, l \in \mathbb{N}\}\}) \cup \mathcal{P}(\{T_{++}, \mathbf{T}_{+\cdot}, \{\mathbf{T}_{\mathcal{X}_l} | \mathcal{X}_l \subseteq \mathcal{X}, l \in \mathbb{N}\}\})$. Note that the unconstrained case of $\mathcal{C}_T = \emptyset$ is handled by the construction in (7). This facilitates sampling $\mathbf{T}^{(1:N)}$ in closed-form and in parallel as shown in step 22 of Alg. 1. The most notable cases of tractable distributions are (see App. B.2.1):

$$\mathbf{T}|\mathbf{\Lambda}, T_{++} \sim \text{Multinomial}(T_{++}, \mathbf{\Lambda}/\Lambda_{++}); \tag{10}$$

$$\mathbf{T}|\mathbf{\Lambda}, \mathbf{T}_{\cdot+} \sim \prod_{i=1}^{I} \text{Multinomial}(T_{i+}, \mathbf{\Lambda}/\Lambda_{i+}); \tag{11}$$

$$\mathbf{T}|\mathbf{\Lambda}, \mathbf{T}_{+\cdot} \sim \prod_{j=1}^{J} \text{Multinomial}(T_{+j}, \mathbf{\Lambda}/\Lambda_{+j}). \tag{12}$$

Each of the distributions above can assimilate cell constraints of the form $\mathcal{C}_T = \{\mathbf{T}_{\mathcal{X}_l} | \mathcal{X}_l \subseteq \mathcal{X}, l \in \mathbb{N}\}$ without violating $\mu$'s tractability by limiting the support of $\mathbf{T}|\mathbf{\Lambda}, \mathcal{C}$. If the constraint set $\mathcal{C}_T$ contains at least both marginals, that is $\mathcal{C}_T^{\text{intractable}} \subseteq \mathcal{P}(\{\mathbf{T}_{\cdot+}, \mathbf{T}_{+\cdot}, \{\mathbf{T}_{\mathcal{X}_l} | \mathcal{X}_l \subseteq \mathcal{X}, l \in \mathbb{N}\}\}) \setminus \mathcal{C}_T^{\text{tractable}}$, then the target distribution becomes Fisher's non-central multivariate hypergeometric [1, 43]:

$$\frac{\prod_{i=1}^{I} T_{i+}! \prod_{j=1}^{J} T_{+j}!}{T_{++}! \prod_{i,j=1}^{I,J} T_{ij}!} \prod_{i,j=1}^{I,J} \left( \frac{\Lambda_{ij}\Lambda_{++}}{\Lambda_{i+}\Lambda_{+j}} \right)^{T_{ij}}. \tag{13}$$

Direct sampling without rejection from this distribution for arbitrary $I, J$ is infeasible [1].

### 3.1.1 Markov Basis MCMC

Therefore, we devise an MCMC proposal on $\mathcal{T}_{\mathcal{C}}$. Using a suite of greedy deterministic Algorithms [6], we can initialise our MCMC with a $\mathbf{T}^{(0)} \in \mathcal{T}_{\mathcal{C}}$. By virtue of definition 3.2 we guarantee that no proposed moves modify the summary statistics in $\mathcal{C}_k$ (Condition 1) and that there exists a path between any two tables such that any table member of the path is $\mathcal{C}_k$-admissible (Condition 2). The collection $\mathcal{C}$ of $K$ constraints generates $K$ sets of Markov bases $\mathcal{M}_1, \ldots, \mathcal{M}_K$. Our proposal mechanism consists of the universe $\mathcal{M} = \bigcap_{k=1}^{K} \mathcal{M}_k$. See App. B.2.2 for info on Markov Bases.

**Definition 3.2.** : A *Markov basis* $\mathcal{M}_k$ is a set of moves $\mathbf{f}_1, \ldots, \mathbf{f}_L : \mathcal{X} \to \mathbb{Z}$ satisfying:

1. $\mathcal{S}_k(\mathbf{f}_l) = \{\mathbf{0}\} \; \forall \; 1 \leq l \leq L = |\mathcal{M}_k|$ and

2. for any two $\mathcal{C}_k$-admissible $\mathbf{T}, \mathbf{T}'$ there are $\mathbf{f}_{l_1}, \ldots, \mathbf{f}_{l_A}$ with $\eta_l \in \mathbb{Z}_{\neq 0}$ such that $\mathbf{T}' = \mathbf{T} + \sum_{m=1}^{A} \eta_l \mathbf{f}_{l_m}$ and $\mathbf{T} + \sum_{m=1}^{a} \eta_l \mathbf{f}_{l_m} \geq 0$ for $1 \leq a \leq A$.

In the case of $I \times J$ ODMs constrained by $\mathcal{C}_T^{\text{intractable}}$, $\mathcal{M}$ consists of functions $\mathbf{f}_1, \ldots, \mathbf{f}_L$ such that $\forall \; x = (i_1, j_1), x' = (i_2, j_2) \in \mathcal{X}$ with $i_1 \neq i_2, j_1 \neq j_2$,

$$\mathbf{f}_l(x) = \begin{cases} \eta & \text{if } x = (i_1, j_1) \text{ or } x = (i_2, j_2); \\ -\eta & \text{if } x = (i_1, j_2) \text{ or } x = (i_2, j_1); \\ 0 & \text{otherwise.} \end{cases} \tag{14}$$

A Gibbs Markov Basis (GMB) sampler can now be constructed (see steps 20 - 22 in Alg. 1).

**Proposition 3.1.** (Adapted from [11]): Let $\mu$ be a probability measure on $\mathcal{T}_{\mathcal{C}}$. Given a Markov basis $\mathcal{M}$ that satisfies 3.2, generate a Markov chain in $\mathcal{T}_{\mathcal{C}}$ by sampling $l$ uniformly at random from $\{1, \ldots, L\}$. Let $\eta \in \mathbb{Z}$. If the chain is at $\mathbf{T} \in \mathcal{T}_{\mathcal{C}}$, determine $\text{supp}(\eta)$ such that $\mathbf{T} + \eta \mathbf{f}_l \geq 0$. Choose

$$\mathbb{P}(\eta) \propto \prod_{x \in \mathcal{X}: \mathbf{f}_l(x) \neq 0} (\mu(T(x) + \eta \mathbf{f}_l(x)))^{-1}$$

and move to $\mathbf{T}' = \mathbf{T} + \eta \mathbf{f}_l$ for the choice of $\eta$. An aperiodic, reversible, connected Markov chain in $\mathcal{T}_{\mathcal{C}}$ is constructed with stationary distribution proportional to $\mu(\mathbf{T})$. Proof is found in [11].

We note that when individual cells are fixed ($\mathbf{T}_{\mathcal{X}_l} \in \mathcal{C}$), the summary statistics operator is applied over $\mathcal{X}_l$ instead of $\mathcal{X}$. Therefore, the size of $\mathcal{M}$ is reduced to comply with Definition 3.2. This may shorten the diameter of the Markov Chain's state space, leading to better mixing times [42].

## 4 Experimental Results

We empirically test our framework on both synthetic and real-world data from Cambridge, UK and Washington, DC, USA. We compare GENSIT against SIM-MCMC [13], SIM-NN [17], SIT-MCMC [47] and the Geo-contextual Multitask Embedding Learner (GMEL) [27]. See App. C.1 for the computational complexities of these methods. GMEL is trained on a larger set of data that includes, besides the cost matrix $\mathbf{C}$, destination-level urban indicators $\mathbf{Y}$, where the log destination attraction $\mathbf{y}$ is a column vector of $\mathbf{Y}$. We validate the generated $\mathbf{T}$ and $\mathbf{\Lambda}$ samples from all methods against the ground truth ODM $\mathbf{T}^*$ using the Standardised Root Mean Square Error (SRMSE) and the 99% high probability region cell coverage probability (CP) (see App. C.2 for definitions). See App. D,E,F for experimental protocols, implementation details, a comprehensive sensitivity study on our framework, and reproducibility using our package `gensit`.

### 4.1 Synthetic ODM Scalability

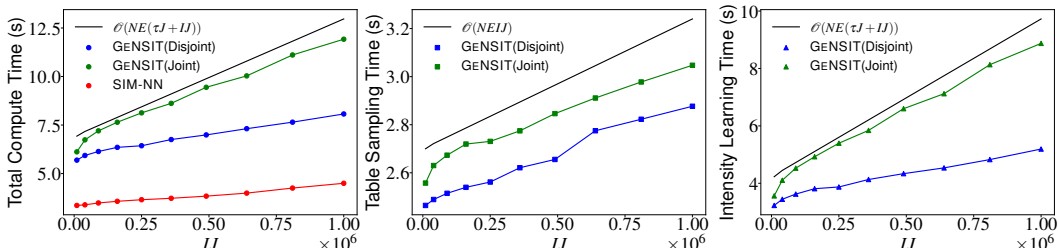

Figure 4: Total computation time (left) of GENSIT and SIM-NN [17] versus the number of origin-destination pairs $IJ$ with $(I \times J)$ equal to $100 \times 100, 200 \times 200, \ldots, 1000 \times 1000$. The two algorithms are run for $N = 10^3$ iterations with $\mathcal{C}_T = \{\mathbf{T}_{.+}, \mathbf{T}_{+.}, \mathbf{T}_{\mathcal{X}_{50\%}}\}$. Constraint $\mathbf{T}_{\mathcal{X}_{50\%}}$ means that 50% of table cells chosen uniformly at random are fixed. Total computation time is the sum of the $\mathbf{T}$ sampling (middle) and $\mathbf{\Lambda}$ learning (right) times. Intensity learning and table sampling times are computed for lines 6-12 and lines 19-24 of Alg. 1, respectively. Our framework scales linearly with $IJ$, which is much faster than SIM-MCMC and SIT-MCMC. SIM-NN does not operate at all in the discrete table space, which explains its faster computational speed.

We begin by offering synthetic experiments for varying ODM dimensions $(I \times J)$ and number of agents $A$, comparing computation time and ground truth reconstruction error (SRMSE) in Figs. 4 and 5. For a fixed $(I \times J)$ dimension we sample an intensity uniformly at random and subsequently generate a ground truth table $\mathbf{T}^*$ with fixed $A = T_{++}$ by sampling from the Multinomial distribution in (10). For Figs. 4 and 5b we set $T_{++} = 10^6$ while for Fig. 5a we vary this for $0.01, 0.1, 0.25, 0.5$ and $1 \times 10^6$.

Our framework achieves a speed-up of $\mathcal{O}(J^2)$ compared to previous methods such as SIM-MCMC, SIT-MCMC.

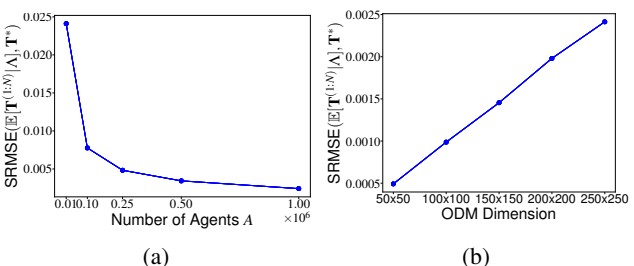

(a)                                    (b)

Figure 5: SRMSE by total number of agents $A$ (left) and ODM dimension $(I \times J)$ (right) of GENSIT's discrete ODM sampling (line 24 of Alg. 1) for $N = 10^4$ iterations, a fixed intensity $\mathbf{\Lambda}$ and $\mathcal{C}_T = \{\mathbf{T}_{.+}, \mathbf{T}_{\mathcal{X}_{50\%}}\}$. On the left we set $(I \times J) = 150 \times 150$. The reconstruction error (SRMSE) scales linearly in $IJ$ and exponentially in $A = T_{++}$.

Reconstruction errors grow linearly in dimension $(I \times J)$ and exponentially in number of agents $A$.

### 4.2 Real-world ODM Reconstructions and Predictions

We now assess our framework's capacity to estimate agent home-to-work trips in real-world ODM prediction problems in Cambridge, UK [47] and Washington DC, USA [27].

#### 4.2.1 Cambridge, UK

In the Cambridge dataset, the ground truth ODM is a $69 \times 13$ contingency table with $33,704$ agents. Tab. 1 shows that reconstruction error (SRMSE) and the % of ground truth cells covered by the 99% high probability region of the ODM samples (CP) are significantly improved when operating in $\mathbf{T}$ level using our Joint scheme compared to SIM-MCMCand SIM-NN, which only operate on the $\mathbf{\Lambda}$ level. We also outperform the competitive SIT-MCMC approach [47], which can operate in the discrete ODM level, since in the limit of $\mathcal{C}_T$ our method can move to high $\mathbf{T}$ probability regions much faster than MCMC due to the optimisation of the SIM $\mathbf{\Lambda}$ parameters. This effectively deflates the $\hat{\mathbf{\Lambda}}$ estimator variance relative to the variance of the $\mathbf{\Lambda}$ samples in MCMC. Only in the absence of rich $\mathcal{C}_T$ data does SIT-MCMC outperform our method (total constrained case) as it limits the support of $\boldsymbol{\theta}$ to $[0,2]^2$ which acts as regularisation. Letting the support of $\boldsymbol{\theta}$ span the entire $\mathbb{R}^2$ allows information carried by stronger $\mathbf{\Lambda}$ constraints (such as $\mathbf{\Lambda}_{+\cdot}$) to permeate to the $\mathbf{T}$ space much faster in GeNSIT compared to SIT-MCMC. The plethora of sources of uncertainty ranging from the HW-SDE (6) to the combinatorial nature of $\mathcal{T}_{\mathcal{C}}$ suggest that GENSIT is faster in reconstructing ground truth ODMs compared to a fully Bayesian approach such as SIT-MCMC.

| ORIGIN-DESTINATION MATRIX | $\mathcal{C}$ | TARGET M | ↓ SRMSE ($\mathbf{M}^{(1:N)}, \mathbf{T}^*$) | | | | | ↑ 99% CP ($\mathbf{M}^{(1:N)}, \mathbf{T}^*$) | | | | |
|---|---|---|---|---|---|---|---|---|---|---|---|---|
| | | | GENSIT DISJOINT | GENSIT JOINT | [SIT-MCMC] | [SIM-NN] | [SIM-MCMC] | GENSIT DISJOINT | GENSIT JOINT | [SIT-MCMC] | [SIM-NN] | [SIM-MCMC] |
| Total constrained | $\mathcal{T}_{++},$ $\mathbf{\Lambda}_{++}$ | $\mathbf{\Lambda}$ | 2.43 | 2.43 | ***0.70*** | 1.08 | ***0.70*** | 23% | 24% | 41% | ***89%*** | 42% |
| | | $\mathbf{T}$ | 2.43 | 2.43 | ***0.70*** | – | – | 30% | 31% | 68% | – | – |
| Singly constrained | $\mathbf{T}_{\cdot+},$ $\mathbf{\Lambda}_{\cdot+}$ | $\mathbf{\Lambda}$ | *0.43* | *0.43* | 0.62 | *0.43* | 0.63 | 90% | 90% | 42% | ***93%*** | 42% |
| | | $\mathbf{T}$ | *0.43* | *0.43* | 0.61 | – | – | **93%** | 93% | 72% | – | – |

$-$: This case is not handled by the approach mentioned in the column.

(a) ODMs with closed-form (tractable) $\mathbf{T}$ distributions (10).

| ORIGIN-DESTINATION MATRIX | $\mathcal{C}$ | TARGET M | ↓ SRMSE ($\mathbf{M}^{(1:N)}, \mathbf{T}^*$) | | | ↑ 99% CP ($\mathbf{M}^{(1:N)}, \mathbf{T}^*$) | | |
|---|---|---|---|---|---|---|---|---|
| | | | GENSIT DISJOINT | GENSIT JOINT | [SIT-MCMC] | GENSIT DISJOINT | GENSIT JOINT | [SIT-MCMC] |
| Doubly constrained | $\mathbf{T}_{\cdot+}, \mathbf{T}_{+\cdot},$ $\mathbf{\Lambda}_{++}$ | $\mathbf{\Lambda}$ | 2.50 | *0.68* | 1.22 | 18% | 78% | 20% |
| | | $\mathbf{T}$ | 1.15 | **0.55** | 0.59 | 68% | **89%** | 86% |
| Doubly and 10% cell constrained | $\mathbf{T}_{\cdot+}, \mathbf{T}_{+\cdot},$ $\mathbf{T}_{\mathcal{X}1}, \mathbf{\Lambda}_{++}$ | $\mathbf{\Lambda}$ | 2.50 | *0.87* | 1.10 | 18% | 79% | 32% |
| | | $\mathbf{T}$ | 1.06 | **0.43** | 0.56 | 71% | **92%** | 88% |
| Doubly and 20% cell constrained | $\mathbf{T}_{\cdot+}, \mathbf{T}_{+\cdot},$ $\mathbf{T}_{\mathcal{X}2}, \mathbf{\Lambda}_{++}$ | $\mathbf{\Lambda}$ | 2.50 | *0.92* | 1.06 | 18% | 78% | 32% |
| | | $\mathbf{T}$ | 1.02 | **0.38** | 0.51 | 74% | **94%** | 90% |

(b) ODMs with intractable $\mathbf{T}$ distribution (13), where conditioning $\mathbf{\Lambda}$ on $\mathcal{C}$ is problematic.

Table 1: Ground truth $\mathbf{T}^*$ validation metrics comparing our method against [13, 17, 47] in the $\mathbf{\Lambda}$ and $\mathbf{T}$ levels across constraint sets $\mathcal{C}$ and $\sigma = 0.141$ (best) for the Cambridge dataset. On a $\mathcal{C}$ basis the best metric in $\mathbf{T},\mathbf{\Lambda}$ spaces is *emphasised* for each of the two and **highlighted** between the two. Inference on the discrete table space offers lower SRMSE and higher CP compared to inference in the continuous space. On an ODM basis we obtain the best reconstruction error (SRMSE) and ground truth coverage (CP) in $\mathbf{T}$-space in all but the totally constrained ODM. This is due to SIT-MCMC forcing $\boldsymbol{\theta} \in [0,2]^2$ instead of $\mathbb{R}^2$, which has a regularisation effect. In the absence of substantial $\mathcal{C}$ data, this effect is more pronounced. See Tab. 6 (App. E.1) for full table across multiple $\sigma$ regimes.

Fig. 6 supports this claim by shedding light on the convergence rate of running mean estimates of the ground truth $\mathbf{T}^*$. The Joint GeNSIT scheme converges to a mean $\mathbf{T}$ estimate much earlier than SIT-MCMC across all $\mathcal{C}$ regimes. Mean $\mathbf{T}$ estimates are also improved in the Joint GeNSIT compared to SIT-MCMC in confined $\mathbf{T}$ spaces (doubly, doubly and 10% cell, doubly and 20% cell constrained ODMs). Under the same $\mathcal{C}$ regimes CP does not improve significantly as $N$ grows large, which suggests that the variance of $\mathbf{T}$ samples appears stable as early as $N = 10^4$. In the Disjoint GeNSIT the information encoded in larger $\mathcal{C}_T$ is not propagated to the $\mathbf{\Lambda}$ updates. As a result, no SRMSE or CP improvements are detected in the course of $N$.

#### 4.2.2 Washington, DC, USA

We apply our method to the Washington dataset, where the ground truth ODM is a $179 \times 179$ contingency table with $200,029$ agents. Tab. 2 reports the reconstruction error (SRMSE) and the % coverage of the ground truth cells (CP) in $\mathbf{\Lambda}$ and $\mathbf{T}$. Comparisons against SIM-MCMCand SIT-MCMCwere infeasible due to their high computational complexity (300 hours to obtain 500 samples on a 32-core NVIDIA GPU). Instead, we leveraged GMEL[27] which operates only in the

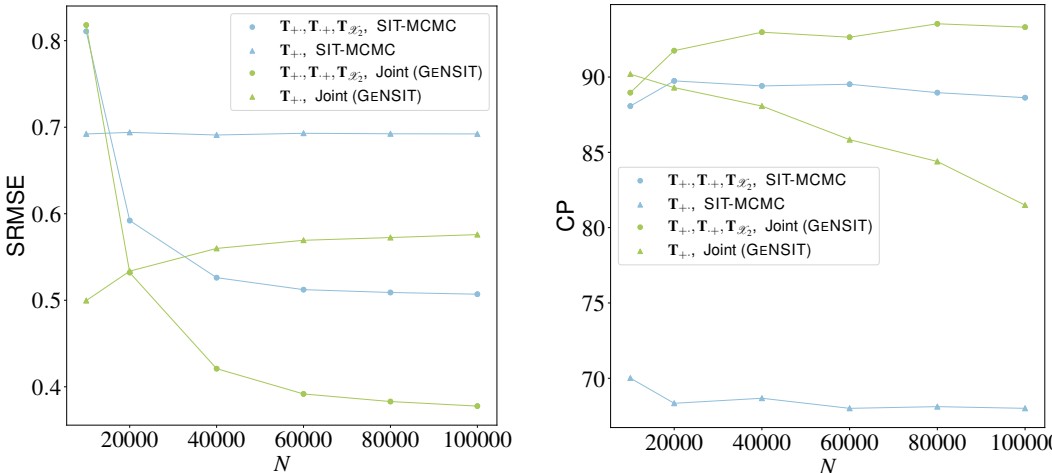

Figure 6: SRMSE and CP for $\mathcal{C} = \{\mathbf{T}_{\cdot+}\}$ ( ● ), $\{\mathbf{T}_{\cdot+}, \mathbf{T}_{+\cdot}, \mathbf{T}_{\mathcal{X}_2}\}$ ( ▲ ) computed cumulatively along $N$ iterations of Alg. 1 for $\mathbf{T}|\mathcal{D}, \mathcal{C}$ samples for GENSIT (Joint) and SIT-MCMC for the Cambridge dataset. The Joint GENSIT converges to a lower SRMSE faster than SIT-MCMC while achieving better $\mathbf{T}^*$ coverage. The singly constrained ODM ($\mathcal{C} = \{\mathbf{T}_{\cdot+}\}$) has very large support $\mathcal{T}_\mathcal{C}$ given that CP is decreasing with N. See App. E for all cases of $\mathcal{C}$ and frameworks.

| FRAMEWORK | DATA $\mathcal{D}$ | TARGET $\mathbf{M}$ | ↓ SRMSE $(\mathbf{M}^{(1:N)}, \mathbf{T}^*)$ | ↑ SSI $(\mathbf{M}^{(1:N)}, \mathbf{T}^*)$ | ↑ 99% CP $(\mathbf{M}^{(1:N)}, \mathbf{T}^*)$ |
|---|---|---|---|---|---|
| [GMEL] | $\mathbf{Y}, \mathbf{C}$ | $\mathbf{\Lambda}$ | $2.43 \pm 0.15$ | $0.38 \pm 0.02$ | $5\% \pm 1\%$ |
| [SIM-NN] | | $\mathbf{\Lambda}$ | $2.47 \pm 0.00$ | $\mathbf{0.51 \pm 0.00}$ | $24\% \pm 3\%$ |
| GENSIT (Disjoint) | $\mathbf{y}, \mathbf{C}$ | $\mathbf{\Lambda}$ | $2.47 \pm 0.00$ | $\mathbf{0.51 \pm 0.00}$ | $19\% \pm 3\%$ |
| | | $\mathbf{T}$ | $2.39 \pm 0.09$ | $0.43 \pm 0.01$ | $\mathbf{47\% \pm 1\%}$ |
| GENSIT (Joint) | | $\mathbf{\Lambda}$ | $2.45 \pm 0.00$ | $0.50 \pm 0.00$ | $2\% \pm 0\%$ |
| | | $\mathbf{T}$ | $\mathbf{2.37 \pm 0.08}$ | $0.45 \pm 0.01$ | $44\% \pm 1\%$ |

Table 2: Ground truth $\mathbf{T}^*$ validation metrics (mean $\pm$ std. for $E = 10$ ensemble size) comparing our method against [27, 17] in the $\mathbf{\Lambda}$ and $\mathbf{T}$ levels for $\mathcal{C} = \{T_{++}, \mathbf{T}_{\mathcal{X}_{\text{train}}}\}$ and $\sigma = 0.141$ (best) for the Washington dataset. Arrow ↑ indicates higher values are better, and vice versa. We achieve the best error (SRMSE) and ground truth coverage (CP) overall (see **bold cells**). The observed data $\mathcal{D}$ leveraged to train GENSIT, SIM-NN is a small subset of the data required to train GMEL ($\mathbf{y}$ is a column vector of $\mathbf{Y}$). See Tab. 7 (App. E.2) for full table across multiple $\sigma$ regimes.

continuous $\mathbf{\Lambda}$ space by learning a mapping between a large feature space $\mathcal{D}$ and $\mathbf{T}_{\mathcal{X}_{\text{train}}}$. Tab. 2 shows that we outperform both GMEL and SIM-NN in terms of reconstruction error and coverage.

## 5   Concluding Remarks

We introduced GeNSIT, an efficient framework for jointly sampling the discrete combinatorial space of agent trips ($\mathbf{T}$) subject to summary statistic data and its continuous mean-field limit $\mathbf{\Lambda}$. We surmount the limitations of methods which operate strictly on $\mathbf{\Lambda}$ space [13, 17, 27] and of methods that incur a large computational cost [47]. We accomplish this by introducing the first framework operating on both $\mathbf{T}, \mathbf{\Lambda}$ that scales linearly with the number of origin-destination pairs $IJ$. We offer enhanced reconstruction error and coverage of the ground truth ODMs in Cambridge, UK and Washington, DC. Although NNs require a much larger number of internal parameters to be calibrated relative to MCMC, their embedding of physics models regularises this parameter space and prevents over-fitting. A remaining open problem on this front is the assimilation of more complex $\mathcal{C}$ structures in agent population synthesis and simulation (see App. B.2.3), since ground truth data is typically partially observed. Our work also relies on the SIM's assumptions about the agents' decision-making process, which in practise is unobserved. An examination of different agent utility models could benefit the applicability of our framework. In terms of our work's social impact, policy decisions made from ABMs of social systems could negatively affect individuals, necessitating expert review and ethics oversight.

## Acknowledgments and Disclosure of Funding

IZ was supported by UK Research and Innovation (UKRI) Research Council and Arup Group PhD studentship. MG was supported by a Royal Academy of Engineering Research Chair grant RC-SRF1718/6/34 and EPSRC grants EP/T000414/1, EP/W005816/1, EP/V056441/1, EP/V056522/1, EP/R018413/2, EP/R034710/1, EP/R004889/1. TD was supported by the UKRI Turing AI acceleration Fellowship EP/V02678X/1. For the purpose of open access, the authors have applied a Creative Commons Attribution (CC-BY) licence to any Author Accepted Manuscript version arising from this submission.

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

# Appendices

**Table of Contents**

# A Nomenclature

We provide a Tab. of notation (see Tab. 3) used throughout the paper.

| Notation | Definition | Domain |
|---|---|---|
| $\mathbb{R}$ | Real numbers. | - |
| $\mathbb{N}$ | Natural numbers including $0$. | - |
| $\mathbb{Z}$ | Integer numbers including $0$. | - |
| $I$ | Number of origin locations | $\mathbb{N}_{>0}$. |
| $J$ | Number of destination locations. | $\mathbb{N}_{>0}$ |
| $A$ | Number of agents modelled. | $\mathbb{N}_{>0}$ |
| $\mathbf{T}$ | Discrete origin-destination matrix (two-way contingency table). | $\mathbb{N}^{I \times J}$ |
| $\mathbf{T}^*$ | Ground truth discrete origin-destination matrix. | $\mathbb{N}^{I \times J}$ |
| $\boldsymbol{\Lambda}$ | Continuous origin-destination matrix. | $\mathbb{R}_{>0}^{I \times J}$ |
| $\mathcal{X}$ | Set of ODM cells whose values are fixed/conditioned. | $\mathcal{P}(\{(1,1), \ldots, (I,J)\})$ |
| $\alpha$ | Spatial interaction model parameter controlling the effect of log destination attractiveness on the expected number of agent trips. | $\mathbb{R}_{>0}$ |
| $\beta$ | Spatial interaction model parameter controlling the effect of travel impedance on the expected number of agent trips. | $\mathbb{R}_{>0}$ |
| $\epsilon$ | Harris-Wilson (HW) SDE responsiveness parameter. | $\mathbb{R}_{>0}$ |
| $\delta$ | HW SDE parameter controlling the smallest number of jobs a destination can have. | $\mathbb{R}_{>0}$ |
| $\kappa$ | HW SDE parameter controlling the number of agents competing for one job. | $\mathbb{R}_{>0}$ |
| $\sigma$ | HW SDE parameter controlling the diffusion term's noise variance. | $\mathbb{R}_{>0}$ |
| $\mathbf{B}_t$ | Wiener process. | $\mathbb{R}^J$ |
| $\boldsymbol{\theta}$ | Parameter vector $(\alpha, \beta)$ learned by the Neural Network. | $\mathbb{R}_{\geq 0}^2$ |
| $\mathbf{z}$ | Destination attraction. | $\mathbb{R}_{\geq 0}^J$ |
| $\mathbf{x}$ | Log destination attraction. | $\mathbb{R}^J$ |
| $\mathbf{y}$ | Observed log destination attraction (log number of jobs). | $\mathbb{R}^J$ |
| $\mathbf{Y}$ | $J \times F$ matrix of destination location features including $\mathbf{x}$ as a column, where $F$ is the total number of destination features. | $\mathbb{R}^{J \times F}$ |
| $\mathbf{C}$ | $I \times J$ cost matrix reflecting agent travel impedance. | $\mathbb{R}_{>0}^{I \times J}$ |
| $\mathbf{D}_{\cdot +}$ | Observed total travel cost by origin location. | $\mathbb{R}_{>0}^I$ |
| $\mathcal{D}$ | Observation data used in the Neural Network's loss function. | $\mathcal{P}(\{\mathbf{y}, \mathbf{D}_{\cdot +}\})$ |
| $\mathbf{T}_{\cdot +}, \mathbf{T}_{\cdot +}$ | Number of agents travelling from each origin (random, deterministic). | $\mathbb{N}_{>0}^I$ |
| $\mathbf{T}_{+ \cdot}, \mathbf{T}_{+ \cdot}$ | Number of agents travelling to each destination (random, deterministic). | $\mathbb{N}_{>0}^J$ |
| $T_{++}$, $T_{++}$ | Total number of agents (random, deterministic). | $\mathbb{N}_{>0}$ |
| $\mathbf{T}_{\mathcal{X}}, \mathbf{T}_{\mathcal{X}}$ | Set of discrete ODM cell values that are fixed/conditioned (random, deterministic). | $\mathbb{N}^{|\mathcal{X}|}$ |
| $\mathcal{C}_T$ | Set of noise-free data constraints in discrete $\mathbf{T}$ space. | $\mathcal{P}(\{\mathbf{T}_{\cdot +}, \mathbf{T}_{+ \cdot}, T_{++}, \mathbf{T}_{\mathcal{X}}\})$ |
| $\boldsymbol{\Lambda}_{\cdot +}$ | Expected number of agents travelling from each origin. | $\mathbb{R}_{>0}^I$ |
| $\boldsymbol{\Lambda}_{+ \cdot}$ | Expected number of agents travelling to each destination. | $\mathbb{R}_{>0}^J$ |
| $\Lambda_{++}$ | Expected total number of agents. | $\mathbb{R}_{>0}$ |
| $\mathcal{C}_{\Lambda}$ | Set of noise-free data constraints in continuous $\boldsymbol{\Lambda}$ space. | $\mathcal{P}(\{\Lambda_{++}, \boldsymbol{\Lambda}_{\cdot +}\})$ or $\mathcal{P}(\{\Lambda_{++}, \boldsymbol{\Lambda}_{+ \cdot}\})$ |
| $\mathcal{C}$ | Set of constraints in discrete $\mathbf{T}$ and continuous $\boldsymbol{\Lambda}$ spaces $\mathcal{C}_T \cup \mathcal{C}_{\Lambda}$. | |
| $\mathcal{T}$ | Space of all non-negative discrete ODMs. | |
| $\mathcal{T}_{\mathcal{C}}$ | Space of all non-negative discrete ODMs constrained by $\mathcal{C}_T$. | |
| $\mathcal{P}$ | Powerset operator. | |
| $\mathcal{S}$ | Summary statistic operator. | |
| $\mathcal{O}$ | Time/space complexity operator. | |
| $\mathbb{P}$ | Probability mass function. | |
| $\mathbb{E}$ | Expectation. | |
| $\odot$ | Hadamard product. | |
| $\mathcal{L}$ | Loss operator using the Neural Network. | |
| $\boldsymbol{\nu}$ | Loss operator hyperparameters $(\sigma_d, \sigma_T, \sigma_{\Lambda})$. | $\mathbb{R}_{>0}^3$. |
| $\psi_{NN}$ | Neural Network forward solver. | - |
| $\mathbf{W}$ | Neural Network weights. | $\mathbb{R}^{(J+1) \times 20 + 21 \times 2}$ |
| $\phi_{HW}$ | Euler-Maruyama numerical solver for the Harris-Wilson SDE. | $\mathbb{R}^J$ |
| $\mathcal{M}$ | Markov Basis moves used in Gibbs sampling of discrete ODMs. | |
| $\mathbf{M}$ | Target ODM $\mathbf{T}$ or $\boldsymbol{\Lambda}$. | |

Table 3: Notation used in the paper.

# B Theory and Methodology Foundations

## B.1 Continuous ODM

### B.1.1 Spatial Interaction Modelling Choice

A fundamental assumption is that the continuous agent trip intensity $\mathbf{\Lambda}$ is governed by a SIM. Therefore, agents are discouraged from travelling long distances due to high travelling costs incurred and encouraged to travel to locations that offer high utility, such as areas with high job availability. This means that highly irregular or non-uniform trip data might not be captured well by $\mathbf{\Lambda}$. However, operating at the discrete ODM level allows us to assimilate trip data as constraints. This shrinks the ODM support and enhances spatial imputation of missing data much better than doing so at the continuous $\mathbf{\Lambda}$ level. In the limit of trip data constraints, we are guaranteed to improve our estimated of missing trip data regardless of their spatial distribution and presence of outliers. This is theoretically founded [11] and empirically shown in Tabs. 1,2 of our Experimental Section. In poorly constrained settings, we rely more on $\mathbf{\Lambda}$ for spatial imputation, which increases reconstruction error. Outlier presence will be reflected in the row/column sum constraints facilitating table sampling in that region of the ODM support. If no row/column sum constraints are imposed, exploring such regions of the support would be significantly hindered.

### B.1.2 Eliciting Agent Utility Functions

In the ODMs considered we have constructed agent utility functions that leverage the log destination attraction at each destination location $\mathbf{x}$ and the total distance (cost) of travelling between any origin and destination $\mathbf{C}$. Additional observations at the origin, destination and origin-destination level can be assimilated by incorporating them as terms in the maximum entropy argument used to derive the functional forms of $\mathbf{\Lambda}$ in equations (3), (4), and (5). The SIM intensity is equivalent to the multinomial logit [30], which allows us to define an arbitrary utility model inside the $\exp$ in the numerator of these equations. For example, one might want to include data $\mathbf{o} \in \mathbb{R}^I$ in addition to the existing $\mathbf{x}, \mathbf{C}$ in the totally constrained SIM (total $= \Lambda_{++}$). Then, we need to maximise:

$$-\sum_{i,j}^{I,J} \Lambda_{ij}(\log(\Lambda_{ij}) - o_i - \alpha x_j + \beta c_{ij}) - \mu(\sum_{ij}^{I,J} \Lambda_{ij} - \Lambda_{++}),$$

where $\mu$ is the Lagrange multiplier. This yields

$$\Lambda_{ij} = \frac{\Lambda_{++}(\exp(o_i + \alpha x_j - \beta c_{ij}))}{\sum_{i,j}^{I,J} \exp(o_i + \alpha x_j - \beta c_{ij})}.$$

## B.2 Discrete ODM

### B.2.1 Target Table Distributions

The target distributions $\mu\left(\mathbf{T} \mid \mathbf{\Lambda}, \mathcal{C}_T\right)$ we sample from in Alg. 1 are listed below in Tab. 4.

We note that the quantity $\omega_{ij} = \frac{\Lambda_{ij}\Lambda_{++}}{\Lambda_{i+}\Lambda_{+j}}$ in the last three rows of Tab. 4 is called the odds ratio and encodes the dependence between rows (origins) and columns (destinations). SIMs incorporate this dependence spatially in the cost matrix $\mathbf{C}$. The case for spatial independence is achieved when $\beta = 0$ in (3), (4). This translates to the travel cost having no effect on the agents' choice of destination. Fisher's non-central multivariate hypergeometric reduces to its central version if and only if $\omega_{ij} = 1$ [1]. The central version corresponds to a uniform distribution over $\mathcal{T}_C$. Additionally, the normalising constant of Fisher's non-central multivariate hypergeometric is a partition function involving a sum over all elements of $\mathcal{T}_C$ satisfying the conditioned margins. By virtue of the extension to Chu-Vandermonde's theorem [5] proved in [47], this normalising constant can be computed in constant time using

$$\prod_j^J \frac{T_{+j}!T_{+j}!}{T_{++}!T_{ij}!} \prod_i^I \left(\frac{\omega_{ij}}{\omega_{+j}}\right)^{T_{ij}}. \tag{15}$$

| ODM | $\mathcal{C}_T$ | $\mu\left(\mathbf{T}\mid\mathbf{\Lambda},\mathcal{C}_T\right)$ | DISTRIBUTION | TRACTABLE |
|---|---|---|---|---|
| Unconstrained | $\emptyset$ | $\prod_{i,j}^{I,J}\left(\frac{\exp(-\Lambda_{ij})\Lambda_{ij}^{T_{ij}}}{T_{ij}!}\right)$ | Poisson | ✓ |
| Totally Constrained | $T_{++}$ | $\prod_{i,j}^{I,J}\left(\frac{T_{++}!}{T_{ij}!}\left(\frac{\Lambda_{ij}}{\Lambda_{++}}\right)^{T_{ij}}\right)$ | Multinomial | ✓ |
| Singly Constrained | $\mathbf{T}_{\cdot+}$ | $\prod_{i,j}^{I,J}\left(\frac{T_{i+}!}{T_{ij}!}\left(\frac{\Lambda_{ij}}{\Lambda_{i+}}\right)^{T_{ij}}\right)$ | Product Multinomial | ✓ |
| Doubly Constrained | $\mathbf{T}_{\cdot+},\mathbf{T}_{+\cdot}$ | $\frac{\prod_{i=1}^{I}T_{i+}!\prod_{j=1}^{J}T_{+j}!}{T_{++}!\prod_{i,j=1}^{I,J}T_{ij}!}\prod_{i,j=1}^{I,J}\left(\frac{\Lambda_{ij}\Lambda_{++}}{\Lambda_{i+}\Lambda_{+j}}\right)^{T_{ij}}$ | Fisher's non-central multivariate hypergeometric | × |
| Doubly Constrained with 10% cells fixed | $\mathbf{T}_{\cdot+},\mathbf{T}_{+\cdot},\mathbf{T}_{\mathcal{X}_1}$ | —"— | —"— | × |
| Doubly Constrained with 20% cells fixed | $\mathbf{T}_{\cdot+},\mathbf{T}_{+\cdot},\mathbf{T}_{\mathcal{X}_2}$ | —"— | —"— | × |

Table 4: List of target $\mathbf{T}$ distributions with their associated constraints $\mathcal{C}_T$.

### B.2.2 Markov Bases Preliminaries

Theoretically analysing the full joint framework is challenging future work, however we offer some additional theoretical insight. We encourage the reader to follow the references of this section for more technical details.

The fundamental theorem of Markov Bases introduced in the seminal work of Diaconis and Sturmfels [11] establishes an equivalence between a Markov Basis and the generator of an ideal of a polynomial ring. By virtue of the Hilbert basis theorem [40], any ideal in a polynomial ring has a finite generating set. Therefore, there exists a *finite* MB for the class of inference problems we are dealing with. This MB connects the fiber, i.e. the support of all discrete ODMs satisfying summary statistics in the form of row, column sums and/or cell constraints. By Proposition 3.1, we can construct a Gibbs MB sampler that will converge to Fisher's non-central hypergeometric distribution on the fiber in finite time.

Moreover, an MB sampler converges to its stationary distribution in at most $A^2$ steps, where $A$ is the total number of agents [46]. Although this result is not directly applicable to our framework in its given form, we conjecture that it can be extended to $\eta=\pm1$ and Fisher's *central* hypergeometric distribution as evidenced by our experimental results. An MB connects tables (elements) of the fiber graph (discrete state-space of tables). The fastest convergence of an MB sampler is achieved by an MB that has the smallest possible diameter, i.e. the longest path between two elements of the fiber graph.

### B.2.3 Incorporating Additional Constraints

Incorporating additional structural constraints remains an open problem. The case for sparse matrices can be handled through the cell constraints $\mathbf{T}_{\mathcal{X}'}$ for $\mathcal{X}'\subseteq\mathcal{X}$. Symmetric ODMs (e.g. an adjacency matrix of a bipartite graph) can be explored as follows. The subspace of $\mathcal{T}_{\mathcal{C}}$ consists of square matrices $I\times I$ with equal row and column sums. We construct an MB on $\mathcal{T}_{\mathcal{C}}$ by applying the following modification to equation (14):

$$\mathbf{f}_l(x)=\begin{cases}+\eta & \text{if } x\in\{(i_1,j_1),(i_2,j_2),(j_1,i_1),(j_2,i_2)\}\\-\eta & \text{if } x\in\{(i_1,j_2),(i_2,j_1),(j_2,i_1),(j_1,i_2)\}\\0 & \text{otherwise.}\end{cases}$$

First, this guarantees that every $2\times2$ move in the original $\mathcal{M}$ that modifies a cell along the diagonal is by definition symmetric as to the other cell it modifies, meaning if cell $(i,j)$ is modified then so is $(j,i)$. Second, without loss of generality assume that a move modifies a $2\times2$ section in the upper triangular section of the ODM. This ensures that the same move is applied symmetrically to the lower triangular section of the ODM. In both cases, any move will guarantee that the ODM will be symmetric after removing any duplicate Markov Bases generated in the process.

### B.3 GENSIT

#### B.3.1 Disjoint versus Joint scheme

The Disjoint scheme consists only of loss terms that depend directly on fully observed data $\mathcal{D}$ (either log-destination attraction $\mathbf{y}$ or total distance agents travelled by origin location $\mathbf{D}_{\cdot+}$). In contrast, the Joint scheme consists of loss terms that either depend on the same fully observed data and on the partially observed table $\mathbf{T}$ through the table marginal $\mathbf{T}|\boldsymbol{\theta}, \mathbf{x}, \mathcal{C}, \mathcal{D}$.

The Joint scheme is an instance of a Gibbs sampler on the full posterior marginals $\boldsymbol{\theta}|(\mathbf{x}, \mathbf{T}, \mathcal{C}, \mathcal{D})$, $\mathbf{x}|(\boldsymbol{\theta}, \mathbf{T}, \mathcal{C}, \mathcal{D})$ and $\mathbf{T}|(\boldsymbol{\theta}, \mathbf{x}, \mathcal{C}, \mathcal{D})$. The Disjoint scheme is an instance of a collapsed Gibbs sampler where we sample from $\boldsymbol{\theta}|(\mathbf{x}, \mathcal{C}, \mathcal{D})$, $\mathbf{x}|(\boldsymbol{\theta}, \mathcal{C}, \mathcal{D})$ and then from $\mathbf{T}|(\boldsymbol{\theta}, \mathbf{x}, \mathcal{C}, \mathcal{D})$. This means we integrate out $\mathbf{T}$ by means of $p(\boldsymbol{\theta}|\mathbf{x}, \mathcal{C}, \mathcal{D}) = \sum_{\mathbf{T}} p(\boldsymbol{\theta}|\mathbf{T}, \mathbf{x}, \mathcal{C}, \mathcal{D}) P(\mathbf{T}|\mathbf{x}, \mathcal{C}, \mathcal{D})$ and $p(\mathbf{x}|\boldsymbol{\theta}, \mathcal{C}, \mathcal{D}) = \sum_{\mathbf{T}} p(\mathbf{x}|\mathbf{T}, \boldsymbol{\theta}, \mathcal{C}, \mathcal{D}) P(\mathbf{T}|\boldsymbol{\theta}, \mathcal{C}, \mathcal{D})$. Therefore, we use the Joint scheme when we have reason to believe that the covariance between $\boldsymbol{\theta}, \mathbf{x}$ and $\mathbf{T}$ is small. This would be the case when the agent trip intensity is influenced by both the Harris-Wilson SDE and the realised number of agent trips. In contrast, we use the Disjoint scheme to accelerate convergence by reducing the covariance between $\boldsymbol{\theta}, \mathbf{x}$ and $\mathbf{T}$. This would be the case when the agent trip intensity is governed only by the Harris-Wilson SDE and not by the realised number of agent trips.

## C  Method Comparisons

In this section, we compare our method's computational complexity to competitive approaches and list the validation metrics employed for empirical comparisons in the main paper.

### C.1  Computational Complexities

Let $N$ be the number of iterations, and $I, J$ be the number of origins, destinations.

1. GENSIT: $\mathcal{O}(NE(\tau J + IJ))$, where $\tau$ is the number of time steps in the Euler-Maruyama solver, $E$ is the ensemble size. We set $\tau = 1$, and $E = 1$.

2. SIM-MCMC [13]: $\mathcal{O}(NJ(LI + J^2))$ (low $\sigma$ regime) and $\mathcal{O}(NIJLKn_p n_t)$ (high $\sigma$ regime), where $L$ is the number of leapfrog steps in Hamiltonian Monte Carlo, $1 < K < N$ is the number of stopping times, and $n_p, n_t$ are the number of particles and temperatures used in Annealed Importance Sampling, respectively. Typical ranges are $n_p \in [10, 100]$, $n_t \in [10, 30]$, and $L \in [1, 10]$ and $E \in [1, 10]$.

3. SIT-MCMC [47]: $\mathcal{O}(NJ(LI + J^2))$ (low noise regime) $\mathcal{O}(NIJLKn_p n_t)$ (high noise regime). See items 1,2 for details.

4. SIM-NN [17]: $\mathcal{O}(NE(\tau J))$. See item 1 for details.

5. GMEL [27]: $\mathcal{O}(n_n n_l((I + J)D + IJ))$ where $\mathbf{Y} \in \mathbb{R}^{(I+J) \times D}$ is the feature matrix for all locations, $D$ is the number of features per location, $n_n < D$ is the number of nodes per layer, and $n_l$ is the number of hidden layers. In the DC dataset, we have $I + J = 179$ since every origin is also a destination (we do not count them twice). Typical ranges include $n_l \in [1, 10]$ and $n_n \in [1, D]$.

### C.2  Validation Metrics

We leverage the following three validations metrics employed throughout the literature [27, 47, 35, 41, 32]. The Standardised Root Mean Square Error (SRMSE) between the target ODM $\mathbf{M}$ (table $\mathbf{T}$ or intensity $\boldsymbol{\Lambda}$) samples and the ground truth $\mathbf{T}^*$ is equal to

$$\text{SRMSE}(\mathbf{M}^{(1:N)}, \mathbf{T}^*) = \sqrt{\frac{\sum_{i,j=1}^{I,J} \left( \mathbb{E}[M_{ij}^{(1:N)}|\mathcal{C}, \mathcal{D}] - T_{ij}^* \right)^2}{IJ}} \left( \frac{\sum_{i,j=1}^{IJ} \mathbb{E}[M_{ij}^{(1:N)}|\mathcal{C}, \mathcal{D}]}{IJ} \right)^{-1},$$

and the Sorensen Similarity Index (SSI) is equal to

$$\text{SSI}(\mathbf{M}^{(1:N)}, \mathbf{T}^*) = \frac{1}{IJ} \sum_{i,j=1}^{I,J} \frac{2 \min \left( \mathbb{E}[M_{ij}^{(1:N)} | \mathcal{C}, \mathcal{D}], T_{ij}^* \right)}{\mathbb{E}[M_{ij}^{(1:N)} | \mathcal{C}, \mathcal{D}] + T_{ij}^*},$$

whose domain is $[0, 1]$ and values closer to 1 imply a better ODM fit. Finally, the coverage probability of the ground truth table $\mathbf{T}^*$ is calculated by first identifying the lower $L_q \left( \mathbf{T}_{ij}^{(1:N)} \right)$ and upper $U_q \left( \mathbf{T}_{ij}^{(1:N)} \right)$ boundaries of the high probability region containing $q\%$ of the total mass for each table cell $x = (i, j) \in \mathcal{X}$. Then, the $q\%$ cell coverage probability is equal to

$$\text{CP}_q(\mathbf{M}^{(1:N)}, \mathbf{T}^*) = \frac{1}{IJ} \sum_{i,j=1}^{IJ} \mathbb{1}\left\{ L_q \left( M_{ij}^{(1:N)} | \mathcal{C}, \mathcal{D} \right) \le T_{ij}^* \le U_q \left( M_{ij}^{(1:N)} | \mathcal{C}, \mathcal{D} \right) \right\}.$$

## D  Experimental Settings

In this section, we detail the implementation details and experimental protocols used to obtain the results in the main paper.

### D.1  Implementation Details

Regarding the HW-SDE, an Euler-Maruyama numerical solver $\phi_{HW}$ is employed throughout the paper with a time discretisation step of $\Delta t = 0.01$ and number of steps $\tau = 1$. The low and high SDE noise levels correspond to $\sigma = 0.014$ and $\sigma = 0.141$, respectively, to establish a comparison against [13, 17, 47]. Following [13, 17, 47], we set the responsiveness parameter $\epsilon = 1$, and the parameter $\delta$ relating to the job availability of a destination where no agents travel to 0, as in [17]. The job competition parameter $\kappa$ is uniquely determined by $\frac{\Lambda_{++} + \delta J}{\sum_{j=1}^{J} \exp(x_j)}$. Moreover, the same transportation network distance-based cost matrix $\mathbf{C}$ as [47] is used.

The set of observation data $\mathcal{D}$ may include the observed log destination attraction $\mathbf{y} \in \mathbb{R}^J$ and the total distance travelled from each origin $\mathbf{D}_{\cdot+} = (\mathbf{T}^* \odot \mathbf{C}) \in \mathbb{R}_{>0}^I$. For Cambridge, both $\mathbf{y}, \mathbf{D}_{\cdot+}$ have been sourced from the UK's population census dataset provided by the Office of National Statistics. Their spatial resolution is regional: middle and lower super output areas for $\mathbf{y}, \mathbf{D}_{\cdot+}$, respectively. For DC, we only have access to a feature matrix $\mathbf{Y} \in \mathbb{R}^{(I+J) \times D}$ from which we extract a column to use as $\mathbf{y}$.

Our NN is a multi-layer perceptron with one hidden layer, implemented in PyTorch [26] and depicted in Fig. 7. The input layer is set to the observed log-destination attractions $\mathbf{y} \in \mathbb{R}^J$ since we are learning the $\boldsymbol{\theta}$ that generates the observed physics $\mathbf{y}$. The output layer is two-dimensional due to the parameter vector $\boldsymbol{\theta} \in \mathbb{R}^2$. For both datasets we set the number of hidden layers to one and number of nodes to 20. The hidden, output layers have a linear and absolute activation functions, respectively. The latter is important in guaranteeing that our $\boldsymbol{\theta}$ estimates are always positive. The NN parameters $\mathbf{W}$ are initialised by sampling uniformly over the region $[0, 4]^{(J+1) \times 20 + 21 \times 2}$.

We use the Adam optimizer [23] with 0.002 learning rate. Bias is initialised uniformly at $[0, 4]$. We follow [13, 47] in fixing $\sigma_d = 0.03$ and $\sigma_T, \sigma_\Lambda$ to 0.07 to

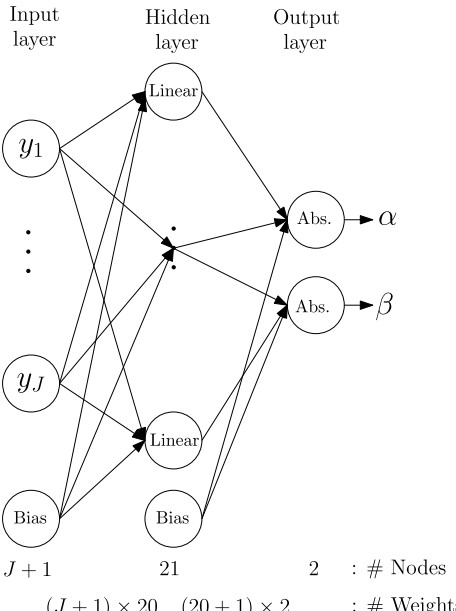

Figure 7: Visual depiction of the Neural Network architecture used in GENSIT for both real-world datasets. The size of the weights $|\mathbf{W}|$ scales linearly in the number of destinations $J$.

reflect a 1% and 3% noise levels, i.e. $\sigma/\log(J) \approx 3\%$. We assume $\mathbf{y}$ are observations from the SDE's stationary distribution, hence our batch size is one. We initialise the Euler-Maruyama solver at $\mathbf{y}$ and run for $\tau = 1$ and step size $\Delta t = 0.001$. At equilibrium the only change in the log-destination attractions is attributed to the SDE's diffusion.

The SDE is encoded in the NN as follows. The Euler-Maruyama solver $\phi_{HW}$ provides forward solutions from $\boldsymbol{\theta}$ to $\mathbf{x}$ at different time steps. The loss function computes the discrepancy between the forward solution $\mathbf{x}$ after $\tau$ steps and the observed data $\mathbf{y}$ at the stationary equilibrium of the SDE, allowing us to encode the physics directly into the NN. A more complicated architecture with a larger number of weights would potentially lead to overfitting the SDE parameters $\boldsymbol{\theta}$ to $\mathbf{y}$ compromising our framework's generalisability.

| NAME | OPERATOR | EVALUATION | $\mathcal{D}$ | $\boldsymbol{\nu}$ | SCHEME |
|---|---|---|---|---|---|
| Negative log destination attraction | $\mathcal{L}(\mathbf{x}\ ;\ \mathcal{D},\boldsymbol{\nu})$ | $\log(\sigma_d) + \frac{1}{2}\sum_{j=1}^{J}\left(\frac{x_j-y_j}{\sigma_d}\right)^2$ | $\mathbf{y}$ | $\sigma_d$ | Disjoint |
| Negative log table | $\mathcal{L}(\mathbf{T},\boldsymbol{\Lambda})$ | $-\log\left(\mu\left(\mathbf{T}/T_{++}|\boldsymbol{\Lambda}/\Lambda_{++},\mathcal{C}\right)\right)$ | $\emptyset$ | - | Joint |
| Total $\mathbf{T}$-based distance travelled by origin | $\mathcal{L}(\mathbf{T}\ ;\ \mathcal{D},\boldsymbol{\nu})$ | $\log(\sigma_d) +$ $\frac{1}{2}\sum_{i=1}^{I}\left(\frac{\sum_{j=1}^{J}\left(T_{ij}c_{ij}-D_{ij}\right)}{\sigma_d}\right)^2$ | $\mathbf{D}_{\cdot+}$ | $\sigma_T$ | Joint |
| Total $\boldsymbol{\Lambda}$-based distance travelled by destination | $\mathcal{L}(\boldsymbol{\Lambda}\ ;\ \mathcal{D},\boldsymbol{\nu})$ | $\log(\sigma_d) +$ $\frac{1}{2}\sum_{i=1}^{I}\left(\frac{\sum_{j=1}^{J}\left(\Lambda_{ij}c_{ij}-D_{ij}\right)}{\sigma_d}\right)^2$ | $\mathbf{D}_{\cdot+}$ | $\sigma_\Lambda$ | Disjoint |
| Joint destination attraction, table loss | $\mathcal{L}(\mathbf{x},\mathbf{T},\boldsymbol{\Lambda}\ ;\ \mathcal{D},\boldsymbol{\nu})$ | $\mathcal{L}(\mathbf{x}\ ;\ \mathcal{D},\boldsymbol{\nu}) + \mathcal{L}(\mathbf{T},\boldsymbol{\Lambda})$ | $\mathbf{y}$ | $\sigma_d$ | Joint |
| Joint destination attraction, total $\mathbf{T}$-based distance loss | $\mathcal{L}(\mathbf{x},\mathbf{T}\ ;\ \mathcal{D},\boldsymbol{\nu})$ | $\mathcal{L}(\mathbf{x}\ ;\ \mathcal{D},\boldsymbol{\nu}) + \mathcal{L}(\mathbf{T}\ ;\ \mathcal{D},\boldsymbol{\nu})$ | $\mathbf{y},\mathbf{D}_{\cdot+}$ | $\sigma_d,\sigma_T$ | Joint |
| Joint destination attraction, total $\boldsymbol{\Lambda}$-based distance loss | $\mathcal{L}(\mathbf{x},\boldsymbol{\Lambda}\ ;\ \mathcal{D},\boldsymbol{\nu})$ | $\mathcal{L}(\mathbf{x}\ ;\ \mathcal{D},\boldsymbol{\nu}) + \mathcal{L}(\boldsymbol{\Lambda}\ ;\ \mathcal{D},\boldsymbol{\nu})$ | $\mathbf{y},\mathbf{D}_{\cdot+}$ | $\sigma_d,\sigma_\Lambda$ | Disjoint |
| Joint table, total $\mathbf{T}$-based distance loss | $\mathcal{L}(\mathbf{T},\boldsymbol{\Lambda}\ ;\ \mathcal{D},\boldsymbol{\nu})$ | $\mathcal{L}(\mathbf{T},\boldsymbol{\Lambda}) + \mathcal{L}(\mathbf{T}\ ;\ \mathcal{D},\boldsymbol{\nu})$ | $\mathbf{D}_{\cdot+}$ | $\sigma_T$ | Joint |
| Joint table, total $\boldsymbol{\Lambda}$-based distance loss | $\mathcal{L}(\mathbf{T},\boldsymbol{\Lambda}\ ;\ \mathcal{D},\boldsymbol{\nu})$ | $\mathcal{L}(\mathbf{T},\boldsymbol{\Lambda}) + \mathcal{L}(\boldsymbol{\Lambda}\ ;\ \mathcal{D},\boldsymbol{\nu})$ | $\mathbf{D}_{\cdot+}$ | $\sigma_\Lambda$ | Joint |
| Joint destination attraction, table, total $\mathbf{T}$-based distance loss | $\mathcal{L}(\mathbf{x},\mathbf{T},\boldsymbol{\Lambda}\ ;\ \mathcal{D},\boldsymbol{\nu})$ | $\mathcal{L}(\mathbf{x}\ ;\ \mathcal{D},\boldsymbol{\nu}) + \mathcal{L}(\mathbf{T},\boldsymbol{\Lambda}) +$ $\mathcal{L}(\mathbf{T}\ ;\ \mathcal{D},\boldsymbol{\nu})$ | $\mathbf{y},\mathbf{D}_{\cdot+}$ | $\sigma_d,\sigma_T$ | Joint |
| Joint destination attraction, table, total $\boldsymbol{\Lambda}$-based distance loss | $\mathcal{L}(\mathbf{x},\mathbf{T},\boldsymbol{\Lambda}\ ;\ \mathcal{D},\boldsymbol{\nu})$ | $\mathcal{L}(\mathbf{x}\ ;\ \mathcal{D},\boldsymbol{\nu}) + \mathcal{L}(\mathbf{T},\boldsymbol{\Lambda}) +$ $\mathcal{L}(\mathbf{T}\ ;\ \mathcal{D},\boldsymbol{\nu})$ | $\mathbf{y},\mathbf{D}_{\cdot+}$ | $\sigma_d,\sigma_\Lambda$ | Joint |

Table 5: List of loss function operators $\mathcal{L}$ and their observation data requirements $\mathcal{D}$.

A list of all loss functions used to train the NN is provided in Tab. 5. The Joint and Disjoint GeNSIT schemes in Figs. 8 and 9 use the loss operators $\mathcal{L}(\mathbf{x}\ ;\ \mathcal{D},\boldsymbol{\nu})$, and $\mathcal{L}(\mathbf{x},\mathbf{T},\boldsymbol{\Lambda}\ ;\ \mathcal{D},\boldsymbol{\nu})$. The observed total distance travelled is assumed to be more noisy than the observed employment data since the latter is a more volatile quantity than the latter and is more prone to errors. We note that SIM-NN in [17] was run under the same NN configuration except for the destination attraction loss $\mathcal{L}(\mathbf{x}\ ;\ \mathcal{D},\boldsymbol{\nu})$ for which the authors used the $L_2$ error. This is equivalent to our loss in Tab. 5 in the limit of $\sigma_d \to 0$. Regarding, table sampling, the observed cells in the Doubly Constrained with 10% cells fixed and Doubly Constrained with 20% cells fixed cases are chosen uniformly at random over the discrete cell support $[1, I] \times [1, J]$. All intractable distributions in Tab. 4 are sampled using Gibbs Markov Basis described in Proposition 3.1. A variant of maximum entropy iterative proportional fitting [6] is employed to initialise tables in Alg. 1.

### D.2 Experimental Protocols

Figures based on synthetic experiments are produced as follows. We generate ground truth tables $(100 \times 100), (200 \times 200), (300 \times 300), (400 \times 400), (500 \times 500), (600 \times 600), (700 \times 700), (800 \times 800), (900 \times 900), (1000 \times 1000)$ as explained in the main paper. For each of these tables, Algorithm 1 and SIM-NN are run for $N = 10^3$, $A = 10^6$, and $\mathcal{C}_T = \{\mathbf{T}_{\cdot+}, \mathbf{T}_{+\cdot}, \mathbf{T}_{\mathcal{X}_{50\%}}\}$ while monitoring the computation times to produce Fig. 4. Additionally, we generate ground truth tables of sizes $(50 \times 50), (100 \times 100), (150 \times 150), (200 \times 200), (250 \times 250)$ with total number of agents $A = 10^6$, $\mathcal{C}_T = \{\mathbf{T}_{\cdot+}, \mathbf{T}_{\mathcal{X}_{50\%}}\}$ and run line 24 of Alg. 1 to produce Fig. 5b. Finally, we generate $150 \times 150$ ground truth tables with $A = 0.01, 0.1, 0.25, 0.5, 1 \times 10^6$ and run the same line of Alg. 1 using $\mathcal{C}_T = \{\mathbf{T}_{\cdot+}, \mathbf{T}_{\mathcal{X}_{50\%}}\}$ and $N = 10^4$ to create Fig. 5a.

Alg. 1 is run for $N = 10^5$ iterations and $E = 1$ ensemble size to produce the data in Figs. 8,10. In Fig. 9 a computational budget of $N \times E = 10^4$ samples is fixed and the following schedule $(N, E)$ is used: $(10, 1000), (50, 200), (100, 100), (500, 20), (1000, 10), (5000, 2), (10000, 1)$. In this case,

estimators of the likes of $\mathbb{E}\left[\cdot \mid \mathbf{T}^*\right]$ are computed over both $N$ and $E$ by coupling all samples across members of the ensemble. The Standardised Root Mean Square Error and $R\%$ High probability region cell Coverage Probability metrics are computed in the same fashion as in [47]. Finally, in Tab. 1 the latent samples $\mathbf{\Lambda}^{(1:N)}, \mathbf{T}^{(1:N)}$ have been trimmed by applying a burning and thinning of 100 and 100 samples across every member of the ensemble.

We follow the setup of [13, 47] to initialise SIM-MCMC , SIT-MCMC and ensure comparability with our models. The computational budget in Tab. 1 and Figs. 10,8 is set to $N = 10^5$. SIM model parameters are fixed to $\epsilon = 1$, $\kappa = 1.025$, $\delta = 0.0128$ and $\sigma_d = 3\% \times \log(J)$. For $\mathbf{x}$ updates the acceptance is monitored to be at least 90%, while the $\boldsymbol{\theta}$-update's acceptance ranges from 30% to 70% depending on the size of the constraint data $\mathcal{C}$. In the high-noise sampling scheme's importance sampling of the normalizing constant, the number of particles is set to 100 and a uniform temperature schedule of 50 inverse temperatures is employed. The percentage of positive signs is maintained around 75% and above.

In the case of the Washington DC data, we employ the same train/test/validation test split as in [27]. In terms of hyperparameter optimisation for GENSIT , we leverage the same architecture as in the Cambridge data and optimise the learning rate on the validation set. To ensure a meaningful comparison, we also optimise the learning rate and multitask weights of the GMEL framework on the validation set. We run each method appearing in Tab. 2 for an ensemble of size $E = 10$ obtain error bars of this ensemble.

All experiments were run using a 32-core CPU machine with 128GB memory. SIM-MCMC and SIT-MCMC took approximately 0.5-1 second per iteration and were for 100 hours in the Cambridge data. SIM-NN took approximately 0.1 seconds per iteration and was run for 50 and 100 hours in the Cambridge, and DC data, respectively. Finally, our framework (GENSIT) took on average 0.2 seconds per iteration and

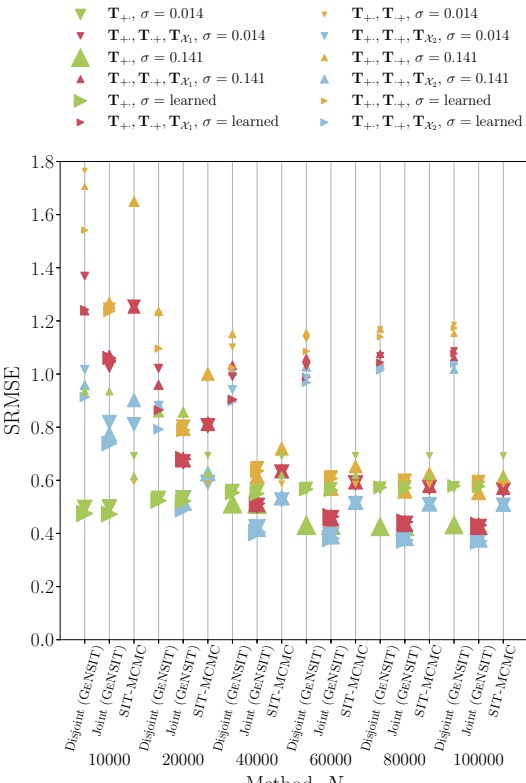

Figure 8: SRMSE (y-axis) and CP ($\propto$ marker size) across $\mathcal{C}$ (marker colour) computed cumulatively along $N$ iterations of Alg. 1 for $\mathbf{T}|\mathcal{D},\mathcal{C}$ samples for GENSIT (Joint and Disjoint) and SIT-MCMC for the Cambridge dataset. The Joint GENSIT converges to a lower SRMSE faster than SIT-MCMC while achieving better $\mathbf{T}^*$ coverage in all but the totally constrained $\mathcal{C}$ cases.

was run for 70 and 120 hours in the Cambridge, and DC data, respectively.

## E    Auxiliary Experimental Results

In this section we provide auxiliary experimental results for the Cambridge and Washington datasets.

### E.1    Cambridge, UK

ODM's reconstruction sensitivity to different ensembles sizes $E$ of Alg. 1 and number of steps $N$ is examined in Fig. 9. A strong preference towards $N \gg E$ is evidenced by significant enhancements in both SRMSE and CP except for the Totally and Singly constrained ODMs (see Tab. 4). All other ODMs employ the GMB sampler 3.1 since the target distribution $\mu$ is intractable. GMB mixes slower than closed-form sampling and therefore needs to be run for a larger number of steps $N$. Despite the fact that a larger ensemble size $E$ facilitates exploration of the non-convex loss landscape, this is not materialised in $\mathbf{T}$-space. The SRMSE's invariance under different $N, E$ configurations detected in

the Totally and Singly ODMs suggests that the conditional independence of successive $\mathbf{T}$ samples compensates for potentially inadequately navigating the loss landscape. Besides, any type of $\mathbf{T}$ sampler used is invariant under initialisation. Therefore, leveraging both optimisation and sampling offers complementary benefits in the estimation of both $\mathbf{\Lambda}$ and $\mathbf{T}$.

The information captured in different $\mathcal{L}$ and $\mathcal{D}$ is portrayed in Fig. 10. Our choice of $\mathcal{L}(\mathbf{x}, \mathbf{T}, \mathbf{\Lambda} \; ; \; \mathcal{D}, \boldsymbol{\nu})$ in the Joint scheme (see 3rd x value from the left) achieves one of the lowest SRMSEs and highest CPs across $\sigma, \mathcal{C}$. Tuning of $\boldsymbol{\nu}$ becomes essential when loss terms assimilating multiple data sources may yield conflicting metrics (lower SRMSE and low CP, and vice versa).

| ORIGIN-DESTINATION MATRIX | $\mathcal{C}$ | $\sigma$ | TARGET M | ↓ SRMSE $(\mathbf{M}^{(1:N)}, \mathbf{T}^*)$ | | | | | ↑ 99% CP $(\mathbf{M}^{(1:N)}, \mathbf{T}^*)$ | | | | |
|---|---|---|---|---|---|---|---|---|---|---|---|---|---|
| | | | | GeNSIT | | [SIT- | [SIM- | [SIM- | GeNSIT | | [SIT- | [SIM- | [SIM- |
| | | | | DISJOINT | JOINT | MCMC] | NN] | MCMC] | DISJOINT | JOINT | MCMC] | NN] | MCMC] |
| Total constrained | $T_{++}$, $\Lambda_{++}$ | 0.014 | $\mathbf{\Lambda}$ | 2.52 | 2.52 | 0.73 | 1.06 | 0.74 | 4% | 4% | 21% | 79% | 21% |
| | | | $\mathbf{T}$ | 2.52 | 2.52 | 0.73 | – | – | 11% | 11% | 67% | – | – |
| | | 0.141 | $\mathbf{\Lambda}$ | 2.43 | 2.43 | ***0.70*** | 1.08 | ***0.70*** | 23% | 24% | 41% | ***89%*** | 42% |
| | | | $\mathbf{T}$ | 2.43 | 2.43 | ***0.70*** | – | – | 30% | 31% | 68% | – | – |
| | | learned | $\mathbf{\Lambda}$ | 2.52 | 2.52 | – | 1.13 | – | 9% | 9% | – | 88% | – |
| | | | $\mathbf{T}$ | 2.52 | 2.52 | – | – | – | 14% | 15% | – | – | – |
| Singly constrained | $T_{.+}$, $\Lambda_{.+}$ | 0.014 | $\mathbf{\Lambda}$ | 0.58 | 0.58 | 0.69 | ***0.43*** | 0.69 | 53% | 53% | 23% | 89% | 23% |
| | | | $\mathbf{T}$ | 0.58 | 0.58 | 0.69 | – | – | 82% | 81% | 68% | – | – |
| | | 0.141 | $\mathbf{\Lambda}$ | ***0.43*** | ***0.43*** | 0.62 | ***0.43*** | 0.63 | 90% | 90% | 42% | ***93%*** | 42% |
| | | | $\mathbf{T}$ | ***0.43*** | ***0.43*** | 0.61 | – | – | ***93%*** | ***93%*** | 72% | – | – |
| | | learned | $\mathbf{\Lambda}$ | 0.58 | 0.58 | – | 0.44 | – | 73% | 72% | – | 88% | – |
| | | | $\mathbf{T}$ | 0.58 | 0.58 | – | – | – | 86% | 87% | – | – | – |

(a) ODMs with closed-form (tractable) $\mathbf{T}$ distributions (10).

| ORIGIN-DESTINATION MATRIX | $\mathcal{C}$ | $\sigma$ | TARGET M | ↓ SRMSE $(\mathbf{M}^{(1:N)}, \mathbf{T}^*)$ | | | ↑ 99% CP $(\mathbf{M}^{(1:N)}, \mathbf{T}^*)$ | | |
|---|---|---|---|---|---|---|---|---|---|
| | | | | GeNSIT | | [SIT-MCMC] | GeNSIT | | [SIT-MCMC] |
| | | | | DISJOINT | JOINT | | DISJOINT | JOINT | |
| Doubly constrained | $\mathbf{T}_{.+}$, $\mathbf{T}_{+.}$, $\Lambda_{++}$ | 0.014 | $\mathbf{\Lambda}$ | 2.52 | *0.67* | 0.71 | 4% | 37% | 21% |
| | | | $\mathbf{T}$ | 1.19 | 0.59 | 0.59 | 61% | 87% | 69% |
| | | 0.141 | $\mathbf{\Lambda}$ | 2.50 | 0.68 | 1.22 | 18% | 78% | 20% |
| | | | $\mathbf{T}$ | 1.15 | ***0.55*** | 0.59 | 68% | ***89%*** | 86% |
| | | learned | $\mathbf{\Lambda}$ | 2.52 | *0.67* | – | 8% | 49% | – |
| | | | $\mathbf{T}$ | 1.17 | 0.59 | – | 66% | 86% | – |
| Doubly and 10% cell constrained | $\mathbf{T}_{.+}$, $\mathbf{T}_{+.}$, $\mathbf{T}x_1$, $\Lambda_{++}$ | 0.014 | $\mathbf{\Lambda}$ | 2.52 | 0.89 | *0.71* | 4% | 49% | 22% |
| | | | $\mathbf{T}$ | 1.09 | ***0.42*** | 0.55 | 69% | ***92%*** | 89% |
| | | 0.141 | $\mathbf{\Lambda}$ | 2.50 | 0.87 | 1.10 | 18% | 79% | 32% |
| | | | $\mathbf{T}$ | 1.06 | 0.43 | 0.56 | 71% | ***92%*** | 88% |
| | | learned | $\mathbf{\Lambda}$ | 2.52 | 0.89 | – | 8% | 51% | – |
| | | | $\mathbf{T}$ | 1.08 | 0.43 | – | 70% | ***92%*** | – |
| Doubly and 20% cell constrained | $\mathbf{T}_{.+}$, $\mathbf{T}_{+.}$, $\mathbf{T}x_2$, $\Lambda_{++}$ | 0.014 | $\mathbf{\Lambda}$ | 2.52 | 0.94 | *0.71* | 4% | 42% | 26% |
| | | | $\mathbf{T}$ | 1.04 | 0.38 | 0.51 | 70% | 92% | 89% |
| | | 0.161 | $\mathbf{\Lambda}$ | 2.50 | 0.92 | 1.06 | 18% | 78% | 32% |
| | | | $\mathbf{T}$ | 1.02 | 0.38 | 0.51 | 74% | ***94%*** | 90% |
| | | learned | $\mathbf{\Lambda}$ | 2.52 | 0.96 | – | 8% | 47% | – |
| | | | $\mathbf{T}$ | 1.04 | ***0.37*** | – | 72% | 93% | – |

$-$: This case is not handled by the approach mentioned in the column.

(b) ODMs with intractable $\mathbf{T}$ distribution (13), where conditioning $\mathbf{\Lambda}$ on $\mathcal{C}$ is problematic.

Table 6: Table 1 expanded to multiple SDE noise $\sigma$ regimes.

Fig. 8 sheds light on the effect of such information propagation on the convergence rate of running estimates of $\mathbb{E}[\cdot \mid \mathcal{D}]$ to the ground truth $\mathbf{T}^*$. The Joint GeNSIT scheme converges to a mean $\mathbf{T}$ estimate much earlier than SIT-MCMC across all $\mathcal{C}$ regimes. Mean $\mathbf{T}$ estimates are improved in the Joint GeNSIT compared to SIT-MCMC in confined $\mathbf{T}$ spaces (Doubly, Doubly and 10% cell, Doubly and 20% cell constrained ODMs). Under the same $\mathcal{C}$ regimes CP does not improve significantly as $N$ grows large, which suggests that the variance of $\mathbf{T}$ samples appears stable as early as $N = 10^4$. In the Disjoint GeNSIT the information encoded in larger $\mathcal{C}_T$ is not propagated to the $\mathbf{\Lambda}$ updates. As a result, no SRMSE or CP improvements are detected in the course of $N$.

### E.2 Washington, DC, USA

We append the expanded Tab. 2 of results for the Washington dataset in Tab. 7.

## F Reproducibility

Our codebase and the real-world data we used are accessible from the Supplementary Material. Trip and employment data for Cambridge, UK are obtained from the Office of National Statistics and used

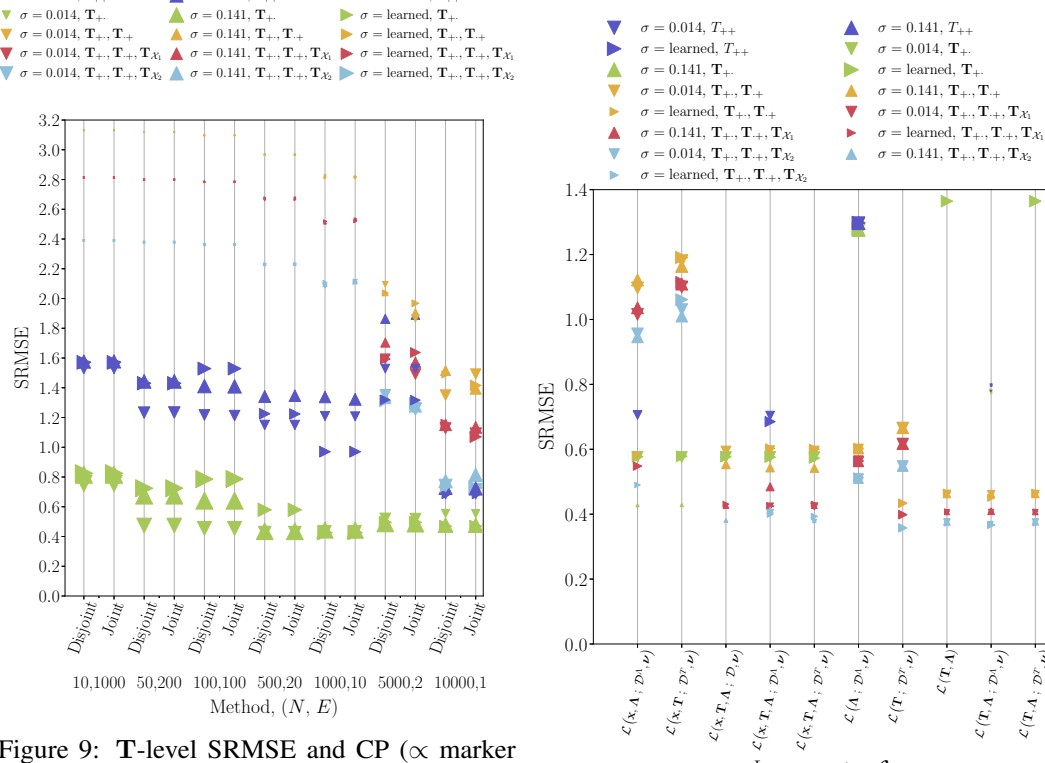

Figure 9: **T**-level SRMSE and CP ($\propto$ marker size) under different combinations of the number of iterations $N$ and ensemble size $E$ used in Alg. 1 for a fixed budget of $N \times E = 10^5$. When $N \gg E$, local $\mathbf{\Lambda}, \mathbf{T}$ information is greedily leveraged compared to global information accumulated over the NN ensemble of $\mathbf{W}^{(0)}$ initialisations. Closed-form **T** sampling eliminates the dependence between successive **T** samples. In contrast, Gibbs Markov Basis sampling requires a larger number of steps $N$ to converge to the stationary target distribution $\mu$ at the expense of higher sample autocorrelation.

Figure 10: **T**-space SRMSE and 99% HMR cell CP ($\propto$ marker size) achieved for different choices of $\mathcal{L}$ (see Tab. 5 for definitions). Both a low SRMSE and a high CP cannot be achieved for these $\mathcal{L}$ choices. This pattern is more prevalent in $\mathcal{L}$'s that assimilate both $\mathbf{y}$ and $\mathbf{D}_{.+}$, suggesting the possibility of contradictory information in the two datasets. Losses that include $\mathcal{L}(\mathbf{T}, \mathbf{\Lambda})$ achieve lower SRMSEs and are not significantly reduced in light of more $\mathcal{C}$ data.

in [47]. The entire feature space for the Washington DC data is accessible through this repository. All data assets leveraged are under the CC-BY 4.0 licence. Once the repository is cloned and a Python virtual environment needs to be created the gensit package can be installed. Depending on the machine capabilities, the number of workers and threads per worker can be set as follows:

```
export $N_WORKERS [input_here]
export $N_THREADS [input_here]
```

### F.1 Cambridge, UK

```
export $CAMBRIDGE cambridge_work_commuter_lsoas_to_msoas
```

The experimental protocols used in the paper can be executed as follows. Experiment 1 can be reproduced by running

| FRAMEWORK | DATA $\mathcal{D}$ | TARGET $\mathbf{M}$ | ↓ SRMSE $(\mathbf{M}^{(1:N)}, \mathbf{T}^*)$ | ↑ SSI $(\mathbf{M}^{(1:N)}, \mathbf{T}^*)$ | ↑ 99% CP $(\mathbf{M}^{(1:N)}, \mathbf{T}^*)$ |
|---|---|---|---|---|---|
| [GMEL] | $\mathbf{Y}, \mathbf{C}$ | $\mathbf{\Lambda}$ | $2.43 \pm 0.15$ | $0.38 \pm 0.02$ | $5\% \pm 1\%$ |
| [SIM-NN] | | $\mathbf{\Lambda}$ | $2.47 \pm 0.00$ (low) $2.47 \pm 0.00$ (high) $2.48 \pm 0.00$ (learned) | $0.50 \pm 0.00$ (low) $\mathbf{0.51 \pm 0.00}$ (high) $0.50 \pm 0.00$ (learned) | $7\% \pm 1\%$ (low) $24\% \pm 3\%$ (high) $0\% \pm 0\%$ (learned) |
| GENSIT (Disjoint) | $\mathbf{y}, \mathbf{C}$ | $\mathbf{\Lambda}$ | $2.47 \pm 0.00$ (low) $2.47 \pm 0.00$ (high) $2.48 \pm 0.00$ (learned) | $0.50 \pm 0.00$ (low) $\mathbf{0.51 \pm 0.00}$ (high) $0.50 \pm 0.00$ (learned) | $6\% \pm 1\%$ (low) $19\% \pm 3\%$ (high) $3\% \pm 1\%$ (learned) |
| | | $\mathbf{T}$ | $2.40 \pm 0.08$ (low) $2.39 \pm 0.09$ (high) $2.40 \pm 0.08$ (learned) | $0.43 \pm 0.01$ (low) $0.43 \pm 0.01$ (high) $0.43 \pm 0.01$ (learned) | $35\% \pm 1\%$ (low) $\mathbf{47\% \pm 1\%}$ (high) $36\% \pm 1\%$ (learned) |
| GENSIT (Joint) | | $\mathbf{\Lambda}$ | $2.45 \pm 0.00$ (low) $2.45 \pm 0.00$ (high) $2.45 \pm 0.00$ (learned) | $0.50 \pm 0.00$ (low) $0.50 \pm 0.00$ (high) $0.50 \pm 0.00$ (learned) | $1\% \pm 0\%$ (low) $2\% \pm 0\%$ (high) $2\% \pm 0\%$ (learned) |
| | | $\mathbf{T}$ | $\mathbf{2.37 \pm 0.08}$ (low) $\mathbf{2.37 \pm 0.08}$ (high) $\mathbf{2.37 \pm 0.09}$ (learned) | $0.45 \pm 0.01$ (low) $0.45 \pm 0.01$ (high) $0.45 \pm 0.01$ (learned) | $44\% \pm 1\%$ (low) $44\% \pm 1\%$ (high) $44\% \pm 1\%$ (learned) |

Table 7: Table 2 expanded to multiple SDE noise $\sigma$ regimes.

```
gensit run ./data/inputs/configs/$CAMBRIDGE/experiment1_disjoint.toml
-sm \
-et SIM_MCMC -et SIM_NN -et NonJointTableSIM_NN \
-nt $N_THREADS -nw $N_WORKERS
```

and

```
gensit run ./data/inputs/configs/$CAMBRIDGE/experiment1_joint.toml
-sm \
-et JointTableSIM_NN \
-nt $N_THREADS -nw $N_WORKERS
```

The exploration-exploitation experiment can be run through commands

```
gensit run ./data/inputs/configs/$CAMBRIDGE/experiment2_disjoint.toml
\
-sm -et SIM_MCMC -et SIM_NN -et NonJointTableSIM_NN \
-nt $N_THREADS -nw $N_WORKERS
```

and

```
gensit run ./data/inputs/configs/$CAMBRIDGE/experiment2_joint.toml \
-sm -et JointTableSIM_NN \
-nt $N_THREADS -nw $N_WORKERS
```

Finally, the comparison of loss functions in experiment 3 can be performed using

```
gensit run ./data/inputs/configs/$CAMBRIDGE/experiment3_joint.toml \
-sm -et JointTableSIM_NN \
-nt $N_THREADS -nw $N_WORKERS
```

## F.2 Washington, DC, USA

The GENSIT Disjoint scheme is run using

```
gensit run ./data/inputs/configs/DC/experiment1_nn_disjoint.toml \
-sm -et NonJointTableSIM_NN \
-nt $N_THREADS -nw $N_WORKERS
```

The GENSIT Joint scheme is run using

```
gensit run ./data/inputs/configs/DC/experiment1_nn_joint.toml \
-sm -et JointTableSIM_NN \
-nt $N_THREADS -nw $N_WORKERS
```

The SIM-NN comparison is run using

```
gensit run ./data/inputs/configs/DC/experiment1_nn_disjoint.toml \
-sm -et SIM_NN \
-nt $N_THREADS -nw $N_WORKERS
```

The GMEL comparison is run using

```
gensit run ./data/inputs/configs/DC/vanilla_comparisons.toml \
-sm -et GraphAttentionNetworkModel_Comparison \
-nt $N_THREADS -nw $N_WORKERS
```

For a detailed explanation of how to reproduce the Tabs. and Figs. we refer the reader to the README.md file in root directory of our codebase, which can be found in the Supplementary Material.

