# OpenReview forum: "Generating Origin-Destination Matrices in Neural Spatial Interaction Models"
_NeurIPS.cc/2024/Conference — NeurIPS 2024 poster_

### Official Review · Reviewer_C8eo · 2024-07-07

**Soundness:** 3
**Presentation:** 2
**Contribution:** 3
**Rating:** 6
**Confidence:** 1

**Summary:**

The paper introduces the GENSIT framework to generate ODMs in agent-based models. The framework addresses challenges in traditional methods that rely on continuous approximations and ad-hoc discretizations, proposing a more efficient approach that operates directly on the discrete combinatorial space. The method leverages neural differential equations to model spatial interactions and demonstrates superior performance in terms of reconstruction error and computational efficiency. The proposed method is illustrated using real-world data from Cambridge, UK, and Washington, DC, USA.

**Strengths:**

1. Combining neural differential equations, the GENSIT framework introduces a novel approach to generating ODMs by directly operating on the discrete space.

2. The framework's effectiveness is demonstrated through empirical validation on serval real-world datasets, showing superior performance in terms of reconstruction error and computational efficiency.

3. The proposed method is computationally efficient and suitable for large-scale datasets.

**Weaknesses:**

1. No theoretical guarantee. I understand that the theoretical aspects of GENSIT may be challenging to analyze. However, I hope the author could provide some theorems on consistency and robustness, which will help the reader better understand this work.

2. Computational cost should be further discussed. The computational complexity is given by $O(NEJ(\tau|W|+I))$, what is the scale of each parameter?

**Questions:**

1. What is the model assumption? Does the GENSIT framework perform well in scenarios where the spatial distribution of the data is highly irregular or non-uniform?

2. Is this method robust to outliers?

3. This method seems very complicated to me. Can you summarize the intuition behind it in a short paragraph?

**Limitations:**

See comments above.

---

> ### Author Rebuttal · Authors · 2024-08-07
>
> We thank the reviewer for their feedback.
> > No theoretical guarantee...
>
> Theoretically analysing the full joint framework is challenging future work, however we offer some existing theoretical insight that we will briefly discuss in the main and expand in the App.
>
> We leverage the universal approximation theorem for analysis the SIM parameter learning using a Neural Net. We also leverage results from the literature on discrete contingency tables complement Proposition 3.1 for the analysis of discrete ODM sampling.
>
> First, the fundamental theorem of Markov Bases [R1] establishes an equivalence between a Markov Basis (MB) and the generator of an ideal of a polynomial ring. By virtue of the Hilbert basis theorem [R3], any ideal in a polynomial ring has a finite generating set. Therefore, there exists a _finite_ MB for the class of inference problems we are dealing with. This MB connects the fiber, i.e. the support of all discrete ODMs satisfying summary statistics in the form of row, column sums and/or cell constraints. By Proposition 3.1, we can construct a Gibbs MB sampler that will converge to Fisher's non-central hypergeometric distribution on the fiber in finite time.
>
> Moreover, we state an important result on the convergence rate of the MB sampler [1,2] in the App.: A MB sampler converges to its stationary distribution in at most $M^2$ steps ($M$: number of agents). Although this result is not directly applicable to us in its given form ($\eta=\pm 1$ on Fisher's central hypergeometric distribution), we conjecture that it can be extended to $\vert\eta\vert>1$ and Fisher's _non-central_ hypergeometric distribution as evidenced by our experimental results. A MB connects tables (elements) of the fiber graph (discrete state-space of tables). The fastest convergence of an MB sampler is achieved by a MB that has the smallest possible diameter (longest path between two elements of the fiber graph).
> > Computational cost should be further discussed...
>
> We point to the global rebuttal (**Theme 3**) where we propose adding a Fig. comparing GeNSIT's computation time with increasing ODM dimensions. We also include this section in the App.:
>
> $N$: number of iterations, $I,J$: number of origins,destinations.
> - GeNSIT: $\mathcal{O}(NE(\tau J\vert+IJ))$, where $\tau$ is the number of time steps in the Euler-Maruyama solver, $E$ is the ensemble size. We set $\tau=1$, and $E=1$.
> - SIM-MCMC [2]: $\mathcal{O}(NJ(LI + J^2))$ (low $\sigma$ regime) and $\mathcal{O}(NIJLKn_pn_t)$ (high $\sigma$ regime), where $L$ is the number of leapfrog steps in Hamiltonian Monte Carlo, $1<K<N$ is the number of stopping times, and $n_p$,$n_t$ are the number of particles and temperatures used in Annealed Importance Sampling, respectively. Typical ranges are $n_p \in [10,100]$, $n_t \in [10,30]$, and $L\in[1,10]$ and $E\in[1,10]$.
> - SIT-MCMC [3]: $\mathcal{O}(NJ(LI + J^2))$ (low noise regime) $\mathcal{O}(NIJLKn_pn_t)$ (high noise regime). See 1,2 for details.
> - SIM-NN [1]: $\mathcal{O}(NE(\tau J))$. See 1. for details.
> - GMEL [4]: $\mathcal{O}(n_nn_l((I+J)D + IJ))$ where $\mathbf{Y} \in \mathbb{R}^{(I+J) \times D}$ is the feature matrix for all locations, $D$ is the number of features per location, $n_n < D$ is the number of nodes per layer, and $n_l$ is the number of hidden layers. In the DC dataset, we have $I+J=179$ since every origin is also a destination (we do not count them twice). Typical ranges include $n_l \in [1,10]$ and $n_n \in [1,D]$.
> > What is the model assumption?...
>
> We briefly mention the following in the main and expand in the App.:
> A fundamental assumption is that the continuous agent trip intensity $\boldsymbol{\Lambda}$ is governed by a SIM. Therefore, agents are discouraged from travelling long distances due to high travelling costs incurred and encouraged to travel to locations that offer high utility, such as areas with high job availability. This means that highly irregular or non-uniform trip data might not be captured well by $\boldsymbol{\Lambda}$. However, operating at the discrete ODM level allows us to assimilate trip data as constraints. This shrinks the ODM support and enhances spatial imputation of missing data much better than doing so at the continuous $\boldsymbol{\Lambda}$ level. In the limit of trip data constraints, we are guaranteed to improve our estimated of missing trip data regardless of their spatial distribution and presence of outliers. This is theoretically founded [R1] and empirically shown in Tabs. 1,2 of our experimental section. In poorly constrained settings, we rely more on $\boldsymbol{\Lambda}$ for spatial imputation, which increases reconstruction error. Outlier presence will be reflected in the row/column sum constraints facilitating table sampling in that region of the ODM support. If no row/column sum constraints are imposed, exploring such regions of the support would be significantly hindered.
> > Can you summarize the intuition behind the method ...?
>
> GeNSIT estimates ODMs summarising agent trip counts between origin and destination locations. It learns how probable it is for agents to travel to a specific destination captured in the continuous trip intensity. This probability is governed by the trade-off between the cost and utility gained from travelling. Conditioned on the intensity, we sample discrete trip counts (ODMs) that satisfy the observed trip summary statistics, such as the agent population at each origin and/or agent demand for a particular destination. Therefore, our framework jointly learns the continuous intensity and discrete ODM until convergence.
>
> # References
>
> [R1]: Diaconis, P., & Sturmfels, B. (1998). Algebraic Algorithms for Sampling from Conditional Distributions. The Annals of Statistics.
>
> [R2]: Windisch, T. (2016). Rapid Mixing and Markov Bases. SIAM Journal on Discrete Mathematics.
>
> [R3]: Simpson, S. G. (1988). Ordinal numbers and the Hilbert basis theorem. The Journal of Symbolic Logic.

---

> > ### Comment · Reviewer_C8eo · 2024-08-09
> > **response to Authors' rebuttal**
> >
> > I appreciate the response, and everything generally makes sense. I will maintain my score for now.

---

### Official Review · Reviewer_h7J4 · 2024-07-07

**Soundness:** 3
**Presentation:** 3
**Contribution:** 3
**Rating:** 6
**Confidence:** 2

**Summary:**

The authors proposed a novel framework that can effectively generate the discrete origin-destination matrices (ODMs) in agent-based models. By using neural differential equations to embed the spatial interactions and operating directly on the discrete combinatorial space, the method overcomes the limitations of traditional continuous models. The superiority of the proposed method is demonstrated in experiments over large-scale real-world datasets.

**Strengths:**

1.	This paper contains many mathematical terms and notations. The authors introduce these notations by giving examples and explanations of their practical meaning, such as the description of the Harris-Wilson system in Eq.6, which improves the quality of the presentation.
2.	In the experiment section, the authors conducted experiments on large-scale datasets that were obtained from real-world cases (Cambridge, UK, and Washington DC, USA). This significantly improves the soundness of this paper.

**Weaknesses:**

1.	The author introduced two sampling schemes in computing the loss for the neural network: the joint and disjoint schemes. This is also described in lines 13 to 15 in Alg.1. However, the description of these two functions is not adequate in the main text; only the difference in the information of T is mentioned. This could be improved if a more detailed explanation of the practical meaning of these two schemes and how they are related to the loss functions is added.
2.	The author claims that the proposed method scales linearly with the number of origin-destination pairs. This major contribution should be mentioned in the experiment section in the main text. Right now, although real-world datasets are used, the experiment part is not easy to understand, and some of the results are not clearly highlighted. The experiment section could be improved if the author gives more results with different scales of data.

**Questions:**

1.	In this work, an ensemble of neural networks is used to predict the vector that governs the utility parameters. I would like to know what networks are used in this part and how different networks affect the reconstruction error.

**Limitations:**

The major limitations are discussed in the experiment section, and no significant negative social impact can be observed in this paper.

---

> ### Author Rebuttal · Authors · 2024-08-07
>
> We thank the reviewer for their helpful comments and important questions that will significantly improve the paper's clarity.
>
> > The author introduced two sampling schemes...
>
> We propose adding the following to the Appendix:
>
> "The Disjoint scheme consists only of loss terms that depend directly on fully observed data $\mathcal{D}$ (either log-destination attraction $\\mathbf{y}$ or total distance agents travelled by origin location $\mathbf{D}\_{\cdot+}$). In contrast, the Joint scheme consists of loss terms that either depend on the same fully observed data and on the partially observed table $\mathbf{T}$ through the table marginal $\mathbf{T}\vert \boldsymbol{\theta}, \mathbf{x},\mathcal{C},\mathcal{D}$.
> The Joint scheme is an instance of a Gibbs sampler on the full posterior marginals $\boldsymbol{\theta}\vert (\mathbf{x},\mathbf{T},\mathcal{C},\mathcal{D})$, $\mathbf{x}\vert (\boldsymbol{\theta}, \mathbf{T},\mathcal{C},\mathcal{D})$ and $\mathbf{T}\vert (\boldsymbol{\theta}, \mathbf{x}, \mathcal{C}, \mathcal{D})$. The Disjoint scheme is an instance of a collapsed Gibbs sampler where we sample from $\boldsymbol{\theta}\vert (\mathbf{x},\mathcal{C},\mathcal{D})$, $\mathbf{x}\vert (\boldsymbol{\theta}, \mathcal{C},\mathcal{D})$ and then from $\mathbf{T}\vert (\boldsymbol{\theta}, \mathbf{x}, \mathcal{C}, \mathcal{D})$. This means we integrate out $\mathbf{T}$ by means of $p(\boldsymbol{\theta}\vert \mathbf{x},\mathcal{C},\mathcal{D}) = \sum_{\mathbf{T}} p(\boldsymbol{\theta}\vert \mathbf{T},\mathbf{x},\mathcal{C},\mathcal{D})P(\mathbf{T}\vert\mathbf{x},\mathcal{C},\mathcal{D})$ and $p(\mathbf{x}\vert \boldsymbol{\theta},\mathcal{C},\mathcal{D}) = \sum_{\mathbf{T}} p(\mathbf{x}\vert \mathbf{T},\boldsymbol{\theta},\mathcal{C},\mathcal{D})P(\mathbf{T}\vert\boldsymbol{\theta},\mathcal{C},\mathcal{D})$.
> Therefore, we use the Joint scheme when we have reason to believe that the covariance between $\boldsymbol{\theta},\mathbf{x}$ and $\mathbf{T}$ is small. This would be the case when the agent trip intensity is influenced by both the Harris-Wilson SDE and the realised number of agent trips. In contrast, we use the Disjoint scheme to accelerate convergence by reducing the covariance between $\boldsymbol{\theta},\mathbf{x}$ and $\mathbf{T}$. This would be the case when the agent trip intensity is governed only by the Harris-Wilson SDE and not by the realised number of agent trips."
>
> We propose expanding the explanation in line 157:
> "The Joint scheme corresponds to a Gibbs sampler on the full posterior marginals $\boldsymbol{\theta}\vert (\mathbf{x},\mathbf{T},\mathcal{C},\mathcal{D})$, $\mathbf{x}\vert (\boldsymbol{\theta}, \mathbf{T},\mathcal{C},\mathcal{D})$ and $\mathbf{T}\vert (\boldsymbol{\theta}, \mathbf{x}, \mathcal{C}, \mathcal{D})$. The Disjoint scheme corresponds to a collapsed Gibbs sampler where we sample from $\boldsymbol{\theta}\vert (\mathbf{x},\mathcal{C},\mathcal{D})$, $\mathbf{x}\vert (\boldsymbol{\theta}, \mathcal{C},\mathcal{D})$ and then from $\mathbf{T}\vert (\boldsymbol{\theta}, \mathbf{x}, \mathcal{C}, \mathcal{D})$ by integrating out $\mathbf{T}$."
>
> To further improve the clarity, we propose adding a column to Table 4 in the Appendix clearly stating whether each loss operator is used in the Joint or Disjoint scheme.
> > The author claims that the proposed method scales linearly...
>
> We appreciate that a lot of results have been presented in the experimental section of the paper. We refer the reviewer to the global rebuttal where we propose ways to improve the clarity of the experimental section (**Theme 2**) as well as include new results on the scalability of GeNSIT across ODM dimensions and number of agents (**Theme 3**).We hope that these Figures will complement the experimental section and highlight that our method scales linearly with the number of origin-destination pairs $IJ$.
> > In this work, an ensemble of neural networks...
>
> Although we include information about the Neural Network (NN) we used in our attached codebase and in lines 434-445 of the Appendix, we recognise that this could be made clearer and more detailed. We refer the reviewer to the global rebuttal (**Theme 2**) where we propose adding two paragraphs in Section C of the Appendix to detail the types of networks and hyper-parameters used in our experiments. A further clarification will be added to the Appendix:
>
> We leverage a multi-layer perceptron with one hidden layer to build a map from the observed log destination attraction $\mathbf{y}$ to the Harris-Wilson SDE parameters $\boldsymbol{\theta}$. The SDE is encoded in the NN as follows. The Euler-Maruyama solver provides forward solutions from $\boldsymbol{\theta}$ to $\mathbf{x}$ at different time steps. The loss function computes the discrepancy between the forward solution $\mathbf{x}$ after $\tau$ steps and the observed data $\mathbf{y}$ at the stationary equilibrium of the SDE, allowing us to encode the physics directly into the NN. A more complicated architecture with a larger number of weights would potentially lead to overfitting the SDE parameters $\boldsymbol{\theta}$ to $\mathbf{y}$ compromising our framework's generalisability.
> > The major limitations are discussed in the experiment section, and no significant negative social impact can be observed in this paper.
>
> We refer the reader to the global rebuttal (**Theme 2**), where this is addressed.

---

### Official Review · Reviewer_dVmA · 2024-07-12

**Soundness:** 3
**Presentation:** 3
**Contribution:** 3
**Rating:** 5
**Confidence:** 5

**Summary:**

This paper introduces a novel framework named Generating Neural Spatial Interaction Tables (GENSIT) for efficiently generating origin-destination (OD) matrices in neural spatial interaction models. The primary objective is to address the challenges of existing methods, such as continuous approximations and ad-hoc discretizations, by operating directly on the discrete combinatorial space of OD matrices. The proposed method utilizes neural differential equations to embed spatial interactions and learns the agents' trip intensity, thereby providing a more accurate and computationally efficient solution. The framework is validated through large-scale spatial mobility agent-based models (ABMs) on multiple real scenarios.

**Strengths:**

- The paper presents a novel approach to generating OD matrices by directly operating in the discrete combinatorial space, which addresses the limitations of continuous approximations. The integration of neural differential equations for embedding spatial interactions is a significant advancement in this domain.
- The framework is thoroughly evaluated using real-world datasets from Cambridge, UK, and Washington, DC, USA. The experimental results demonstrate that GENSIT outperforms existing methods in terms of reconstruction error and ground truth matrix coverage, while also being computationally efficient.
- The paper is well-structured and clearly presents the motivation, methodology, and results. Figures and tables are effectively used to illustrate the performance improvements of the proposed method.

**Weaknesses:**

- The paper does not sufficiently explain the practical importance of generating OD matrices efficiently. For readers unfamiliar with this line of literature, it would be beneficial to provide more context on why efficiency in this process matters and how it impacts the overall utility of ABMs in real-world applications.
- While the methodology is appealing, some parts of the explanation, particularly the notations used in the paper, could be simplified for better understanding. Including more intuitive explanations or examples might help in making the content more accessible.

**Questions:**

- Could you provide more details on why generating OD matrices efficiently is crucial in practice? Specifically, how does this efficiency impact the usability and effectiveness of ABMs in policy-making and other applications?
- The paper mentions that GENSIT outperforms prior art in terms of reconstruction error and computational cost. It would be helpful to include a detailed comparison with these methods, highlighting the specific scenarios where GENSIT provides the most significant advantages.

---

> ### Author Rebuttal · Authors · 2024-08-07
>
> We thank the reviewer for their comments and questions. These will improve the paper's clarity and accessibility to a wider audience.
>
> > The paper does not sufficiently explain the practical importance of generating OD matrices efficiently. For readers unfamiliar with this line of literature, it would be beneficial to provide more context on why efficiency in this process matters and how it impacts the overall utility of ABMs in real-world applications.
>
> We appreciate that the paper would significantly benefit from a more elaborate explanation of the importance of generating ODMs efficiently and the impact this has on ABM simulation and calibration. We refer the reviewer to the global rebuttal (**Theme 1**) where we propose elaborating on exactly this.
>
> > While the methodology is appealing, some parts of the explanation, particularly the notations used in the paper, could be simplified for better understanding. Including more intuitive explanations or examples might help in making the content more accessible.
>
> Thank you. We refer the reviewer to the global rebuttal (**Theme 2**) where we suggest clarity modifications relevant to the reviewer's comment. We also offer additional experimental results and analysis in the global rebuttal (**Theme 3**) to make the content more accessible. If the reviewer has additional quantities in mind in need of further intuition please let us know.
>
> > Could you provide more details on why generating OD matrices efficiently is crucial in practice? Specifically, how does this efficiency impact the usability and effectiveness of ABMs in policy-making and other applications?
>
> We refer the reviewer to the global rebuttal (**Theme 1**) where we propose elaborating on exactly this in our introduction.
>
> > The paper mentions that GENSIT outperforms prior art in terms of reconstruction error and computational cost. It would be helpful to include a detailed comparison with these methods, highlighting the specific scenarios where GENSIT provides the most significant advantages.
>
> We now better highlight the specific scenarios of most significant advantage of GenSIT by explicitly describing these settings in the start of the experimental section (see global rebuttal **Theme 2**), in the analysis of the experiments and also in the concluding remarks where we also highlight settings where GenSIT advantages are limited (totally constrained ODM). The computational cost comparison is detailed by additional experiments listed in the global rebuttal (**Theme 3**) as well as a new section in the Appendix listing all methods' computational complexities (see rebuttal to reviewer C8eo).

---

> > ### Comment · Reviewer_dVmA · 2024-08-14
> > **Thanks for the response**
> >
> > Thanks for the response. I will keep my original score.

---

### Official Review · Reviewer_qQL1 · 2024-07-12

**Soundness:** 2
**Presentation:** 2
**Contribution:** 2
**Rating:** 5
**Confidence:** 1

**Summary:**

The authors propose a generative model for origin-destination matrices $I \times J$ with known marginals. The model assumes that the counts in the matrix are realizations of Poisson random variables conditioned on parameters determined by an unknown intensity $\Lambda$. The matrix entries must also respect the marginal constraints of this unknown intensity. Additionally, they must satisfy a model parameterized by two parameters, $\alpha$ and $\beta$, which control the attractiveness of each destination and the costs between pairs of origin-destination locations.

**Strengths:**

GeNSIT introduces an efficient framework for jointly sampling both discrete combinatorial space of agent trips (T) and its continuous mean-field limit ($\Lambda$). This dual approach is a novel contribution. Another strength is that the framework scales linearly with the number of origin-destination pairs (I+J). GeNSIT regularizes the parameter space of neural networks by embedding physics models. This integration of physics models regularizes the model adding reliability and robustness to the framework.

**Weaknesses:**

I found this article very difficult to understand. One reason is that it seems to have been written for experts in origin-destination matrices, which is a very small group of people at this conference. A more important reason is that the paper is unnecessarily theoretical, overusing mathematical notation and jargon, and leaving several points unclear, which hinders comprehension. Despite the authors' rigor in definitions and notation, there appear to be errors or inaccuracies in several parts that left me very confused.

To be more specific, consider the constraints discussed in (3), (4), and (5). These pertain to the marginals $\Lambda_{++}, \Lambda_{i+}, \lambda_{+j}$  of the intensity $Lambda$. Since this intensity is unknown, these models cannot be trained. Later on, the authors seem to state that they will replace the marginals based on $Lambda$ with the sufficient statistics $T_{++}, T_{i+}, T_{+j}$. If this is the case, why isn't the problem presented directly in terms of these statistics? Or will the marginals of the function $\Lambda$ to be learned satisfy, for example, $\Lambda_{i+} = T_{i+}$?

The authors aim for precision and rigor, but the text falls short and is confusing. Here are some examples:

- The attractiveness of destination $z_j$ is governed by a stochastic differential equation (SDE)? But your algorithm outputs a single $ Z = (z_1, ..., z_J) $ array. Is this single array an asymptotic result of the SDE?
- In equation (7), make it explicit that this is the distribution conditioned on $\Lambda_{ij}$ and that you assume the counts $ T_{ij} $ are independent conditionally. This is mentioned in passing a few lines below.
- Why is $\mathcal{C}_{\Lambda} \subset \mathcal{C}_T$? We have that $\mathcal{C}_{\Lambda}$ is the set of distributions with a constraint on $\Lambda$ and  $\mathcal{C}_T$ is the set of tables realized with fixed margins. How can one be contained within the other if the sets pertain to different elements (distributions indexed by intensity in one case, and count tables in the other)?
- In practice, how can $\mathcal{C}_{\Lambda}$ be known to be used as input in the algorithm?
- What happens if the inputs $\mathcal{C}_{\Lambda}$ are completely inconsistent with the inputs $\mathcal{C}_T$? For example, $C_{++} = 20000$ and $\Lambda_{++} = 100$?
- In equation (9), what is $ k $? Did I miss the definition of this notation? I keep going back through the text to check if I missed a definition.
- "Upon conditioning on sufficient summary statistics $\mathcal{C}_T$, any choice of probability model in (7) becomes equivalent [1]." I don't understand what "equivalent" means here. Do you mean that, conditioned on the sufficient statistics, the data becomes independent of the parameters and there is no way to choose among the different realizations ${T}$ that meet the constraints?
- "The hard-coded constraints $\mathcal{C}_T$ are treated as noise-free data on the discrete table space." How can they be free of random noise? They are sums of Poisson distributions and therefore have a Poisson distribution (for example, $T_{i+} \sim \text{Poisson}(\Lambda_{+i})$).
- "The aforementioned random variables are summary statistics uniquely determined upon conditioning on $T$." This doesn’t make sense. Conditioning on the entire matrix $T_{ij}$, the marginals become degenerate random variables with a single value with probability 1.

**Questions:**

The real data sets are relatively small (69x13 and 179x179). Can you run large synthetic examples to study the scalability of GeNSIT?

**Limitations:**

They addressed adequately the limitations.

---

> ### Author Rebuttal · Authors · 2024-08-07
>
> We thank the reviewer for their feedback.
> > I found this...
>
> Please see **Themes 1,3** in global rebuttal.
> > To be more specific, consider...
>
> The SIM's intensities in (3),(4) are embedded in the Harris-Wilson SDE. The goal is to solve the SDE's inverse problem by learning the parameters $\boldsymbol{\theta} = (\alpha,\beta)$ appearing in (3),(4). The intensity $\boldsymbol{\Lambda}$ marginals will indeed be fixed to satisfy $\Lambda_{++} = \mathbb{E}[T\_{++}\vert \mathcal{C}\_T]$ and $\boldsymbol{\Lambda}\_{\cdot+} = \mathbb{E}[\mathbf{T}\_{\cdot+}\vert \mathcal{C}\_T]$ when $T\_{++},\mathbf{T}\_{\cdot+} \in \mathcal{C}\_T$. So, $\mathbf{T}$-level constraints will match $\boldsymbol{\Lambda}$-level constraints except for the case in (5). This is because the doubly constrained SIM is non-identifiable as argued in line 114.
> > The attractiveness of destination $z_j$ is...
>
> The $\mathbf{z}$ is the numerical solution of the SDE after $\tau$ steps, which we aim to explicitly state it in the exposition of SIMs.
> > In equation (7), make it explicit that...
>
> We replace equation (7) with $T_{ij} \vert \Lambda_{ij} \sim \text{Poisson}(\Lambda_{ij})$ and extend line 131 to "... number of agents travelling from $i$ to $j$, and $T_{ij} \perp T_{i'j'} \vert \Lambda_{ij},\Lambda_{i'j'} \;\forall\; i\neq i', j\neq j'$".
> > Why is $\mathcal{C}_{\Lambda} \subseteq \mathcal{C}_T$?...
>
> We replace $\mathcal{C}\_{\Lambda} \subseteq \mathcal{C}\_T$ with the explanation that "every summary statistic constraint in $\Lambda$-space is also applied in $\mathbf{T}$-space, i.e. we always set $\Lambda\_{++} = \mathbb{E}[T\_{++}\vert \mathcal{C}\_T]$ and $\Lambda_{i+} = \mathbb{E}[T\_{i+}\vert \mathcal{C}\_T]$".
> > ...how can $\mathcal{C}_{\Lambda}$ be...
>
> In practise, we choose $\mathcal{C}\_{\Lambda}$ between $\{\Lambda\_{++}\}$ and $\{ \Lambda\_{++},\boldsymbol{\Lambda}\_{\cdot+} \}$. When $\mathcal{C}\_T = \{T\_{++},\mathbf{T}\_{\cdot+}\}$, we set $\mathcal{C}\_{\Lambda} = \{\Lambda\_{++},\boldsymbol{\Lambda}\_{\cdot+}\}$. In all other cases we set $\mathcal{C}\_{\Lambda}= \{\Lambda\_{++}\}$. When $\mathcal{C}\_{T}$ contains at least $T\_{++},\mathbf{T}_{\cdot+},\mathbf{T}\_{+\cdot}$ and $\mathcal{C}\_{\Lambda} = \{\Lambda\_{++},\boldsymbol{\Lambda}\_{\cdot+}\}$, the dependence between $\mathbf{T}\vert \boldsymbol{\Lambda},\mathcal{C}$ and $\mathbf{y}\vert \mathbf{x}, \mathcal{C}$. As a result, constraints are implicitly weighted (hard $\mathcal{C}\_{\Lambda}$ and soft $\mathcal{C}\_T$ constraints), which inflicts identifiability issues in $\boldsymbol{\Lambda}$.
> > What happens if the inputs $\mathcal{C}\_{\Lambda}$ are...
>
> We always fix $\Lambda_{++} = \mathbb{E}[T\_{++}\vert \mathcal{C}\_T]$ and $\Lambda\_{i+} = \mathbb{E}[T\_{i+}\vert \mathcal{C}\_T]$, so we would not encounter such a case. Despite this, the $\boldsymbol{\Lambda}$ information is propagated to and from $\mathbf{T}$ via $\frac{\boldsymbol{\Lambda}}{\Lambda\_{++}}$ in $\mathbf{T}\vert \boldsymbol{\Lambda},\mathcal{C}_T$.
> > In equation (9), what is $k$?...
>
> Many thanks for picking up our omission. Index $k$ refers to the $k$-th basis operator applied over the cell subset $\mathcal{X}\_k$. We will explicitly state this in line 173.
> > Upon conditioning on sufficient summary statistics $\mathcal{C}\_T$...
>
> We recognise that our writing needs improvement here. The Poisson model for $T\_{ij}\vert \Lambda\_{ij}$ is a modelling choice of the practitioner. Another potential choice would have been $T_{ij} \sim \text{Binomial}(\Lambda_{++},\frac{\Lambda\_{ij}}{\Lambda\_{++}})$. By conditioning on the summary statistics $T\_{++},T\_{i+},T\_{+j}$ these two modelling choices would both yield the same conditional distributions $\mathbf{T}\vert \boldsymbol{\Lambda},\mathcal{C}\_T$. In that sense the two modelling choices are equivalent under conditioning on $\mathcal{C}\_T$. We will clarify this in the main text.
> > "The hard-coded constraints $\mathcal{C}_T$ are treated as...
>
> We apologise for the confusion here. $\mathcal{C}\_T$ are realisations of the Poisson random variables $T\_{++}\vert\boldsymbol{\Lambda},T\_{i+}\vert\boldsymbol{\Lambda},T\_{+j}\vert\boldsymbol{\Lambda}$. By conditioning on $\mathcal{C}\_T$, these variables are no longer random. We propose explicitly stating this in line 135.
> > "The aforementioned random variables...
>
> We propose replacing our sentence with "The aforementioned summary statistics become Dirac random variables upon conditioning on $\mathbf{T}$."
> > The real data sets are relatively small...
>
> We argue that modelling between 900 (69x13) to 32k (179x179) cells and approximately 40k-100k agents are sufficiently large datasets given that they are used to model entire cities such as Cambridge, UK and DC, USA.
>
> We also point the reviewer to the global rebuttal (**Theme 3**), where we introduce new experimental results on GeNSIT's scalability and propose to include them in the main and Appendix.
> > This is a theoretical paper with no immediate connection to societal impact
>
> We'd like to respectfully argue against this statement and provide evidence to the contrary. This is not just a theoretical paper: it introduces methodologies and algorithms that are applied to real-world datasets and ABMs of up to 32,000 cell dimensions and 100,000 agents. We also refer the reviewer to the global rebuttal (**Theme 1**) where we propose elaborating on the benefits of ODM estimation to ABM calibration as well as the social impact of our work.
>
>  # References
>
> [R1]: Ferguson, N. et. al (2006). Strategies for mitigating an influenza pandemic. Nature.
>
> [R2]: Ferguson, Neil et. al (2020). Report 9: Impact of non-pharmaceutical interventions (NPIs) to reduce COVID19 mortality and healthcare demand. Imperial College London.
>
> [R3]: W Axhausen, Kay, Andreas Horni, and Kai Nagel (2016). The multi-agent transport simulation MATSim. Ubiquity Press.
>
> [R4]: Crooks, A., et. al (2021). Agent-Based Modeling and the City: A Gallery of Applications. Urban Informatics.

---

> > ### Comment · Reviewer_qQL1 · 2024-08-13
> >
> > Thanks for responding to my questions. Based on the reply, I have increased my score.

---

### Official Review · Reviewer_C3cp · 2024-07-21

**Soundness:** 3
**Presentation:** 2
**Contribution:** 3
**Rating:** 6
**Confidence:** 3

**Summary:**

The paper considers the task of estimating discrete Origin-Destination Matrices (ODM) which will be useful for generating synthetic agent populations. Rather than the computationally inefficient approach of searching the huge space of such matrices using expensive Agent-based Simulations, the study takes the approach of generating continuous approximations of ODMs (intensity matrices) using a Spatial Interaction Model, whose parameters are estimated using a neural network. Then the continuous ODMs are used to estimate the discrete ODMs through Gibbs Markov Basis Sampling.

**Strengths:**

The main strengths of the paper are as follows:
1) It links the existing literature on spatial interaction models and the Harris-Wilson system with agent-based mobility simulation
2) The work provides a solid approach to estimating the OD intensity matrices from observed data using the SIM
3) Further, it provides a new approach to solve the combinatorial problem of finding the discrete ODM, based on Gibbs Markov Basis Sampling
4) They validate their approach with good experimental results

**Weaknesses:**

I would flag two main concerns:
1) The way this paper has been written, its contributions seem to be too specific to the task of generating discrete OD matrices, which may be useful to a specific application (agent-based population synthesis). I think the general approach (especially estimating parameters of the SIM, sampling discrete matrices using Gibbs Markov Basis Sampling) etc have merit beyond this task, but somehow that is not coming through
2) I did not find the paper very easy to read. I think a table with notations will be useful, also a bit of more clarity on Spatial Interaction Models will be useful. Also, the nature of the observations (y, D) is not very clear.

**Questions:**

1) What are the observations that we have, based on which we are calibrating the SIM using neural network? Is it only the attractiveness of each location? Can there be other versions of the problem where more observations are available?
2) I understand that the task of estimating the discrete ODM is a combinatorial one, but can't we significantly reduce the space by imposing realistic constraints on the matrix structure?

---

> ### Author Rebuttal · Authors · 2024-08-07
>
> We thank the reviewer for their feedback.
> > ..contributions seem to be too specific to the task of generating discrete OD matrices...
>
> We point to the global rebuttal (**Theme 1**) where we elaborate on our framework's wider applicability, importance, connection to ABMs, and societal impact.
> > ...I think a table with notations will be useful...
>
> Please refer to **Theme 2** in the global rebuttal where we include a notation Tab.
>
> To improve the exposition of SIMs, we add the following before line 119 to link SIMs to the multinomial logit:
>
> "We note that additional data at the origin, destination and origin-destination level can be assimilated into SIMs. This can be achieved by incorporating them as terms in the maximum entropy argument used to derive the $\boldsymbol{\Lambda}$ functional forms in equations (3), (4), and (5). We note that the SIM's $\boldsymbol{\Lambda}$ is equivalent to the multinomial logit [5], which generalises our $\boldsymbol{\Lambda}$ construction to accommodate for more data."
>
> We explain this in the App.:
>
> The set of observation data $\mathcal{D}$ may include the observed log destination attraction $\mathbf{y}\in \mathbb{R}^{J}$ and the total distance travelled from each origin $\mathbf{D}\_{\cdot+} = (\mathbf{T}^{*} \odot \mathbf{C}) \in \mathbb{R}\_{>0}^{I}$. For Cambridge, both $\mathbf{y}$, $\mathbf{D}$ have been sourced from the UK's population census dataset provided by the Office of National Statistics. Their spatial resolution is regional: middle and lower super output areas for $\mathbf{y}$, $\mathcal{D}$, respectively. For DC, we only have access to a feature matrix $\mathbf{Y} \in \mathbb{R}^{(I+J)\times D}$ from which we extract a column to use as $\mathbf{y}$.
> > What are the observations that we have, based on which we are calibrating the SIM using neural network?...
>
> We leverage the observed log destination attraction at each destination location. Our sensitivity study in Fig. 6 p.14 also leverages the total distance travelled by agents from each origin location $\mathbf{D}\_{\cdot+} = (\mathbf{T}^{*} \odot \mathbf{C})$. Also, our Joint scheme's loss on $\mathbf{T}\vert \boldsymbol{\Lambda},\mathcal{C}_T,\mathcal{D}$ propagates the information from the summary statistic data $\mathcal{C}_T$ to $\boldsymbol{\Lambda}$. The SDE solution is expressed in terms of the log origin/destination attraction we need observations on the solution to calibrate the SIM parameters using the Neural Net.
>
> Additional observations at the origin, destination and origin-destination level can assimilated by incorporating them as terms in the maximum entropy argument used to derive the functional forms of $\boldsymbol{\Lambda}$ in equations (3), (4), and (5). The SIM intensity is equivalent to the multinomial logit [R1], which allows us to define an arbitrary utility model inside the $\exp$ in the numerator of these equations. For example, one might want to include data $\mathbf{o} \in \mathbb{R}^{I}$ in addition to the existing $\mathbf{x}, \mathbf{C}$ in the totally constrained SIM (total $=M$). Then, we need to maximise:
> $$-\sum_{i,j}^{I,J} \Lambda_{ij}(\log(\Lambda_{ij}) -o_i-\alpha x_j+\beta c_{ij}) - \mu(\sum_{ij}^{I,J} \Lambda_{ij} - M),$$
> where $\mu$ is the Lagrange multiplier. This yields $$\Lambda_{ij} = \frac{M(\exp(o_i+\alpha x_j-\beta c_{ij}))}{\sum_{i,j}^{I,J} \exp(o_i+\alpha x_j - \beta c_{ij})}.$$
> We propose adding a paragraph in a section in the App. as guidance to practitioners who wish to construct new utility functions.
> > ...can we significantly reduce the space by imposing realistic constraints on the matrix structure?
>
> We are exploring the space of discrete ODMs $\mathbf{T}$ with summary statistics $\mathcal{C}\_T$ known as the fiber $\mathcal{T}\_{\mathcal{C}}$ using a finite Markov Basis (MB). Additional structural constraints on the ODM is equivalent to finding a subspace of $\mathcal{T}\_{\mathcal{C}}$ where these structural constraints are satisfied. For this, we update our MB to explore this subspace without breaking the connectivity of the fiber graph (i.e. irreducibility).
>
> Sparse matrices can be handled through the cell constraints $\mathbf{T}\_{\mathcal{X}'}$ for $\mathcal{X}'\subseteq\mathcal{X}$. Symmetric ODMs (e.g. an adjacency matrix of a bipartite graph) can be explored as follows. The subspace of $\mathcal{T}\_{\mathcal{C}}$ consists of square matrices $I\times I$ with equal row and column sums. We construct a MB on $\mathcal{T}\_{\mathcal{C}}$ by applying the following modification to equation (14) (see **Note**):
> $$\mathbf{f}_l(x) = \begin{cases}
>     +\eta & \text{if } x \in \{(i_1,j_1),(i_2,j_2),(j_1,i_1),(j_2,i_2)\} \\
>     -\eta  & \text{if } x \in \{(i_1,j_2),(i_2,j_1),(j_2,i_1),(j_1,i_2)\} \\
>     0 & \text{otherwise.}
> \end{cases}$$
> First, this guarantees that every $2\times 2$ move in the original $\mathcal{M}$ that modifies a cell along the diagonal is by definition symmetric as to the other cell it modifies, meaning if cell $(i,j)$ is modified then so is $(j,i)$. Second, without loss of generality assume that a move modifies a $2\times2$ section in the upper triangular section of the ODM. This ensures that the same move is applied symmetrically to the lower triangular section of the ODM. In both cases, any move will guarantee that the ODM will be symmetric after removing any duplicate Markov Bases generated in the process.
>
> We propose detailing the case for symmetric ODMs in the App. In general, encoding structural constraints on the ODM remains an open challenge, and propose stating this in our conclusion.
>
> **Note**: Equation (14) should read:
> $$\mathbf{f}_l(x) = \begin{cases}
>     +\eta & \text{if } x \in \{(i_1,j_1),(i_2,j_2)\} \\\
>     -\eta  & \text{if } x \in \{(i_1,j_2),(i_2,j_1)\} \\\
>     0 & \text{otherwise.}
> \end{cases}$$
> # References
> [R1]: Nijkamp, P., & Reggiani, A. (1988). Entropy, Spatial Interaction Models and Discrete Choice Analysis: Static and Dynamic Analogies. European Journal of Operational Research.

---

> > ### Comment · Reviewer_C3cp · 2024-08-10
> >
> > I thank the authors for their responses. Accordingly, I have updated my score.

---

### Official Review · Reviewer_sr5H · 2024-07-29

**Soundness:** 3
**Presentation:** 3
**Contribution:** 3
**Rating:** 6
**Confidence:** 3

**Summary:**

The paper introduces a new framework, GENSIT, for calculating origin-destination matrices for Agent-Based Models (ABMs) representing the movement of individual agents, from partially observed summary statistics. The method uses a neural differential equation as a physics-based way of embedding spatial interactions. This allows benefits over prior methods such as scaling linearly with the number of origin-destination pairs IJ. The authors demonstrate improved compute performance and accuracy over prior methods on two datasets of Cambridge, UK and Washington D.C, US.

**Strengths:**

The paper provides a novel contribution to methods for estimating origin-destination matrices, using neural differential equations. It provides a conceptual motivation for this; incorporating a physics based model which matches the domain, and also shows experimental evidence of improvement over prior techniques.

The paper appears high quality. It is well written with no typos that I noticed. There are clear diagrams to illustrate the method, and there is comprehensive explanation of the mathematics, with equations provided in detail.

The paper is significant to reducing the computational cost of estimating origin-destination matrices for Agent-based models, which is a sub-problem of Agent-based modelling.

**Weaknesses:**

Despite being well written, the paper is often dense and a bit tricky to understand. It is heavy with mathematics and often uses technical terms without explaining what they mean in this context (e.g. _"explore the multimodal matrix distribution over a discrete combinatorial support, and incurs discretisation errors"_ in the paper abstract.)

The experimental results seem good, however the table they are presented in was complicated, and hard to understand at first glance what the improvement is over existing techniques, and what each of the columns of the table actually mean. It might be worth having a summary table which only shows the high level results in a clear way.

The authors could perhaps give more context of how this method fits into Agent-based Modelling overall, and how it could be used by ABM practitioners.

The paper is generally well referenced, however there is no mention or explanation of the connection to the wider field of ABM calibration (i.e. estimating unknown parameters for ABMs), which is broader than just Origin-destination matrix estimation. It would be good to add some mention of this broader context of ABM calibration, the challenges involved, and how this technique helps.  The paper claims to reduce the cost of estimating origin-destination matrices, however it's not clear whether that is a computationally limiting step for ABMs, is this a large fraction of the overall compute required for ABMs? What practical benefit is there to having more accurate and faster calculation of Origin-destination matrices?

The paper does not give much information on the implementation details of the neural network for the neural differential equation. There is a reference for PyTorch, but it is not stated explicitly whether PyTorch is used. There are no values for hyperparameters and the architecture of the neural network is not described. It is not specified which optimisation algorithm is used for training.

**Questions:**

The algorithm is quite complex and there are many different parts to the algorithm described (Algorithm 1 in the paper). It is unclear which parts are the most important for the improved performance claimed by the paper, since there is no ablation study of the individual components of the algorithm.

Would an ablation study be feasible? Or is there a reason that all the parts of the algorithm must be included together?

**Limitations:**

The paper mentions that the problem of more complex cell structure constraints for population synthesis remains unsolved.

There is a potential limitation that it may be hard to get accurate ground truth of people movement data in real life scenarios to validate methods such as this. This is not mentioned in the paper (as far as I could see).

The paper could mention the potential risks when simulating social behaviour that the behaviour is oversimplified and doesn't capture the nuances of real human behaviour, or that decisions made on the basis of agent-based models of social systems could negatively affect some individuals.

---

> ### Author Rebuttal · Authors · 2024-08-07
>
> We thank the reviewer for their feedback.
> > Despite being well written, the paper is often dense and ...
>
> Please see global rebuttal (**Theme 2**), where we list improvements we now make to increase clarity throughout, improve notation, and provide a more gentle introduction to technical concepts. We welcome additional suggestions.
> > The experimental results seem good, however...
>
> We refer the reviewer to the global rebuttal (**Theme 2**) where we address all reviewers' comments on improving the presentation of experimental results.
> > The authors could perhaps give more context of how this method fits into Agent-based Modelling overall ... The paper is generally well referenced, however...
>
> We refer the reader to the global rebuttal (**Theme 1**) where we propose adding two paragraphs in the introduction of the main paper to clarify the motivation, link to ABMs and the impact of our framework.
> > The paper does not include much information on the implementation details...
>
> Although we include these details in our codebase and briefly described them in our App. (lines 434-445), we refer the reviewer to the global rebuttal (**Theme 2**) where we propose updating the App. to include an expanded description of the neural network implementation details including the architecture, optimisation algorithm, and hyper-parameters. We also improve the link to that information in the main paper and provide a diagram of the architecture in Fig.2 of the pdf in the global rebuttal.
> > The algorithm is quite complex and ... Would an ablation study be feasible?...
>
> We argue that we are effectively performing a component-based ablation study through our comparison with SIM-MCMC [R2], SIT-MCMC [R3], SIM-NN [R1] and propose that we explicitly mention this in our experimental section. SIM-MCMC and SIT-MCMC both use a Markov Chain Monte Carlo (MCMC) sampler to obtain $\boldsymbol{\theta}$ estimates from the Harris-Wilson SDE. This corresponds to replacing lines 6-17 of Algorithm 1 with an MCMC sampler. Tables 1 and 2 clearly indicate that our method and SIM-NN [R1], which both leverage a Neural Network to learn $\boldsymbol{\theta}$ and a Euler-Maruyama solver to sample $\mathbf{x}\vert\boldsymbol{\theta}$, are superior to the MCMC-based methods in terms of reconstruction error and coverage of the ground truth ODM.  Moreover, neither SIM-NN nor SIM-MCMC sample discrete ODMs as GeNSIT does in lines 18-23 of Algorithm 1. Therefore, comparing against these methods allows us to test the impact that operating only in the continuous intensity $\boldsymbol{\Lambda}$ space has in the overall reconstruction error and coverage of the ground truth ODM. Tables 1 and 2 demonstrate a significant improvement in these two metrics when ODM estimation is performed in the discrete $\mathbf{T}$ space. Therefore, we can conclude that the components of GeNSIT are superior to those found in the prior art because of the computational savings and improved $\boldsymbol{\theta}$ estimates brought by the Neural Network (lines 6-17 of Algorithm 1) and the enhanced reconstruction and coverage brought by discrete ODM sampling (lines 18-23 of Algorithm 1).
>
> Apart from this comparison, we have now run additional synthetic experiments provided in the global rebuttal (**Theme 3**) where we perform a component-based ablation study of the computation time and SRMSE of each component along increasing ODM dimensions $(I,J)$ (Fig. 2) as well as a study of the discrete ODM sampling component for varying ODM dimensions (Fig. 3) and number of agents $M$ (Fig. 4). We believe these new synthetic experiments will complement the ablation study provided in the experimental section of the paper and propose including Fig 2 in the introduction of the experimental section and Figs. 3 and 4 in Section C of the App.
> > There is a potential limitation that...
>
> Indeed, such data would not be generally accessible. This is precisely why we need a robust framework for estimating origin-destination matrices when fully observed ground truth data is not available, such as GeNSIT. We have leveraged two datasets where ground truth data exists in order to be able to validate GeNSIT. Moreover, it is often the case when summary statistic data are readily available (e.g. agent population at a regional level) from population censuses. Our method can flexibly assimilate these types of data as noise-free constraints in the discrete ODM space. We propose identifying this limitation in our concluding remarks.
> > The paper could mention the potential risks...
>
> We propose to re-emphasize the point on the potential negative impact of policy interventions elicited from ABM situations, and refer the reviewer to the global rebuttal (**Theme 1**) where we propose extending our introduction to discuss amongst other things the social impact of ABMs and GeNSIT's limitations.
>
> # References
>
> [R1]: Gaskin, T., Pavliotis, G. A., & Girolami, M. (2023). Neural Parameter Calibration for Large-Scale Multiagent Models. Proceedings of the National Academy of Sciences.
>
> [R2]: Ellam, L., Girolami, M., Pavliotis, G. A., & Wilson, A. (2018). Stochastic Modelling of Urban Structure. Proceedings of the Royal Society A: Mathematical, Physical and Engineering Sciences.
>
> [R3]: Zachos, I., Damoulas, T., & Girolami, M. (2023). Table Inference for Combinatorial Origin-Destination Choices in Agent-Based Population Synthesis. Stat.

---

> ### Comment · Reviewer_sr5H · 2024-08-12
> **Acknowledgement of rebuttal and thanks**
>
> I thank the authors for providing this detailed rebuttal that addresses all my points. In the global rebuttal it seems you have made many improvements to the paper that cover all my suggested improvements. I think this definitely improves the paper on the dimensions of clarity and presentation in particular.
>
> I have updated my 'presentation' score to 3: good

---

### Author Rebuttal · Authors · 2024-08-07

We thank all reviewers. All agree on the paper's contributions: "novel contribution" [sr5H,qQL1,dVmA,h7J4,C8eo], "new approach" [C3cp]. The "paper appears high quality ... is significant" [sr5H], "provides a solid approach" [C3cp], is "well-structured and clearly presents the motivation" [dVmA]. An "efficient" [qQL1,dVmA,C8eo] framework adds "reliability", "robustness" [qQL1] and "soundness" [h7J4] with a "thorough evaluation" [dVmA] and "good experimental results" [C3cp], which demonstrate "superior performance" [C8eo], "outperforming existing methods" [dVmA].

We now address feedback on motivation (**Theme 1**), clarity (**Theme 2**), and scalability (**Theme 3**).
# Theme 1: Motivation (sr5H, dVmA, C3cp, qQL1)

We add at our intro:
> "Our framework has merit beyond ODM sampling in ABMs. The challenge of learning discrete contingency tables constrained by their summary statistics extends to other fields. Contingency tables have been widely studied in multiple instance learning [27, 11, 39] and ecological inference [31, 30, 32]. In Neuroscience one estimates the efficiency, cost and resilience (equivalent to $T_{ij}$) of neural pathways between pairs of brain regions $(i,j)$ to understand communication and information processing [R2,R3]. Epidemiology also investigates social contact matrices quantifying the number of contacts ($T_{ij}$) between types of individuals $(i,j)$ stratified by demographic characteristics, such as age [R4]."

We append to line 20:
> "This is achieved by computationally expensive forward simulations, which hinders ABM parameter calibration and large-scale testing of multiple policy scenarios [R6]."

We expand line 35:
> "It is also necessary for population synthesis in ABMs [14], which is performed prior to simulation in order to reduce the size of the ABM's parameter space.  Moreover, it avoids ...".

We append to line 61:
> "This approximation effectively acts as a cheap ABM surrogate or emulator, facilitating faster forward simulations to be run by practitioners [R5]. This has tanglible benefits to ABM calibration allowing faster exploration of the parameter space. Our enhanced ODM reconstruction error demonstrates GeNSIT's ability to sufficiently approximate the ABM simulator at a fraction of the computational cost.".

We discuss limitations and social impact in conclusions:
> "Our work also relies on the SIM's assumptions about the agents' decision-making process, which in practise is unobserved. An examination of different agent utility models could benefit the applicability of our framework. In terms of our work's social impact, policy decisions made from ABMs of social systems could negatively affect individuals, necessitating expert review and ethics oversight.".
# Theme 2: Clarity (sr5H, qQL1, C3cp, dVmA)
We now provide a Table of Notation, attached pdf (Tab. 1).

App. Section C now gives full details on the NN. The architecture in Fig. 1 in pdf and lines 434-445 read:
> "Our NN is a multi-layer perceptron with one hidden layer, implemented in PyTorch [R1]. The input layer is set to the observed log-destination attractions $\mathbf{y}\in \mathbb{R}^{J}$ since we are learning the $\boldsymbol{\theta}$ that generates the observed physics $\mathbf{y}$. The output layer is two-dimensional due to the parameter vector $\boldsymbol{\theta}\in\mathbb{R}^2$. For both datasets we set the number of hidden layers to one and number of nodes to 20. The hidden/output layer has a linear/absolute activation function.
>
> We use the Adam optimizer with $0.002$ learning rate. Bias is initialised uniformly at $[0,4]$. We follow [1,2] in fixing $\sigma_d=0.03$ and $\sigma_{T},\sigma_{\Lambda}$ to $0.07$ to reflect a $1\%$ and $3\%$ noise levels, i.e. $\sigma / \log(J) \approx 3\%$. We assume $\mathbf{y}$ are observations from the SDE's stationary distribution, hence our batch size is one. We initialise the Euler-Maruyama solver at $\mathbf{y}$ and run for $\tau=1$ and step size $\Delta t=0.001$. At equilibrium the only change in the log-destination attractions is attributed to the SDE's diffusion."

In Tabs. 1 and 2 we rename columns to SRMSE$(\mathbf{M}^{(1:N)},\mathbf{T}^{\*})$, CP$(\mathbf{M}^{(1:N)},\mathbf{T}^{\*})$, and SSI$(\mathbf{M}^{(1:N)},\mathbf{T}^{\*})$ and link to their exact formulation in the App.

For Tab.  1, we keep the best SRMSE, CP for each method and ODM across all noise regimes, and move the rest to App., Section B.

For Tab.  2, we keep the high-noise results and move the rest to App., Section B. We explain the second column (FEATs. $\mathcal{D}$) in the caption.
# Theme 3: Scalability (qQL1, h7J4, C8eo, dVmA)
We now offer synthetic experiments (attached pdf) for varying ODM dimensions $(I,J)$ and number of agents $M$, comparing computation time and ground truth reconstruction error (SRMSE).

Our complexity is linear in $IJ$. We rederive the complexity fully in terms of $J$ and without $\vert\mathbf{W}\vert$. Our complexity is $\mathcal{O}(NE(\tau J + IJ))$ and we include Fig.2 (pdf), in the experimental section showing how our method's table sampling, intensity learning and total computation times scale with the ODM dimensions. Figs. 3-4  showing the variation of SRMSE are discussed in the main and App.

# References
[R1]: Paszke, A. et.al. (2019). Pytorch: An imperative style, high-performance deep learning library. NeurIPS.

[R2]: Seguin, C., et.al. (2023). Brain Network Communication: Concepts, Models and Applications. Nature Reviews Neuroscience.

[R3]: Goñi, J., et. al (2014). Resting-Brain Functional Connectivity Predicted by Analytic Measures of Network Communication. PNAS.

[R4]: Mistry, D., et. al (2021). Inferring High-Resolution Human Mixing Patterns for Disease Modeling. Nature Communications.

[R5]: Banks, D. L., & Hooten, M. B. (2021). Statistical Challenges in Agent-Based Modeling. The American Statistician.

[R6]: Lempert, R. (2002). Agent-based modeling as organizational and public policy simulators. PNAS.

---

### Author Response · Authors · 2024-08-13
**Acknowledgement**

Dear Reviewers,

Thank you for your reviews and taking the time to respond to our rebuttal. If you have any additional comments or requests we would be happy to address them in the remaining reviewer-author discussion period. If you believe we have already addressed your corrections/questions we would appreciate if this could be reflected in your updated score.

Many Thanks,
Authors

---

### Decision · Program_Chairs · 2024-09-25

**Decision:**

Accept (poster)

**Comment:**

The paper discusses the estimation of so-called origin-destination matrices in agent-based models.

The paper is generally well received by the reviewers and is written to a high standard, providing a lot of mathematical details. Reviewers also positively commented about the application to real data.  However, the method described is relevant only to a relatively small subarea of research.

A minor issue is that the referencing could generally be improved. As some reviewers have pointed out, the connection to the field of calibrating ABMs is not really discussed. Similarly the field of Bayesian calibration of computer models, there is a broad literature discussing with posteriors of the form of eq (8), see e.g. [1 on page 4, 2, equation (3), 3 just to give some examples].

[1] Ridgway, J. (2017). Probably approximate Bayesian computation: nonasymptotic convergence of ABC under misspecification. arXiv preprint arXiv:1707.05987.

[2] Schmon, S. M., Cannon, P. W., & Knoblauch, J. (2020). Generalized posteriors in approximate Bayesian computation. arXiv preprint arXiv:2011.08644.

[3] Park, M., Jitkrittum, W., & Sejdinovic, D. (2016, May). K2-ABC: Approximate Bayesian computation with kernel embeddings. In Artificial intelligence and statistics (pp. 398-407). PMLR.